# Structure and function of axo-axonic inhibition

Casey M Schneider-Mizell[1]*, Agnes L Bodor[1†], Forrest Collman[1†], Derrick Brittain[1†], Adam Bleckert[1], Sven Dorkenwald[2,3†], Nicholas L Turner[2,3†], Thomas Macrina[2,3†], Kisuk Lee[2,4†], Ran Lu[2†], Jingpeng Wu[2†], Jun Zhuang[1], Anirban Nandi[1], Brian Hu[1], JoAnn Buchanan[1], Marc M Takeno[1], Russel Torres[1], Gayathri Mahalingam[1], Daniel J Bumbarger[1], Yang Li[1], Thomas Chartrand[1], Nico Kemnitz[2], William M Silversmith[2], Dodam Ih[2], Jonathan Zung[2], Aleksandar Zlateski[2], Ignacio Tartavull[2], Sergiy Popovych[2,3], William Wong[2], Manuel Castro[2], Chris S Jordan[2], Emmanouil Froudarakis[5,6], Lynne Becker[1], Shelby Suckow[1], Jacob Reimer[5,6], Andreas S Tolias[5,6,7], Costas A Anastassiou[1,8‡], H Sebastian Seung[2,3], R Clay Reid[1], Nuno Maçarico da Costa[1]*

[1]Allen Institute for Brain Sciences, Seattle, United States; [2]Princeton Neuroscience Institute, Princeton University, Princeton, United States; [3]Computer Science Department, Princeton University, Princeton, United States; [4]Brain & Cognitive Sciences Department, Massachusetts Institute of Technology, Cambridge, United States; [5]Department of Neuroscience, Baylor College of Medicine, Houston, United States; [6]Center for Neuroscience and Artificial Intelligence, Baylor College of Medicine, Houston, United States; [7]Department of Electrical and Computer Engineering, Rice University, Houston, United States; [8]Department of Neurology, University of British Columbia, Vancouver, Canada

*For correspondence:
caseys@alleninstitute.org
(CMS-M);
nunod@alleninstitute.org
(NMdC)

†These authors contributed equally to this work

Present address: ‡Department of Neurosurgery, Department of Neurology and Regenerative Medicine Institute, Los Angeles, United States

**Abstract** Inhibitory neurons in mammalian cortex exhibit diverse physiological, morphological, molecular, and connectivity signatures. While considerable work has measured the average connectivity of several interneuron classes, there remains a fundamental lack of understanding of the connectivity distribution of distinct inhibitory cell types with synaptic resolution, how it relates to properties of target cells, and how it affects function. Here, we used large-scale electron microscopy and functional imaging to address these questions for chandelier cells in layer 2/3 of the mouse visual cortex. With dense reconstructions from electron microscopy, we mapped the complete chandelier input onto 153 pyramidal neurons. We found that synapse number is highly variable across the population and is correlated with several structural features of the target neuron. This variability in the number of axo-axonic ChC synapses is higher than the variability seen in perisomatic inhibition. Biophysical simulations show that the observed pattern of axo-axonic inhibition is particularly effective in controlling excitatory output when excitation and inhibition are co-active. Finally, we measured chandelier cell activity in awake animals using a cell-type-specific calcium imaging approach and saw highly correlated activity across chandelier cells. In the same experiments, in vivo chandelier population activity correlated with pupil dilation, a proxy for arousal. Together, these results suggest that chandelier cells provide a circuit-wide signal whose strength is adjusted relative to the properties of target neurons.

## Editor's evaluation

This paper will be of high interest to a broad audience of neuroscientists as it provides a major advancement of our understanding of cortical circuits. The quality and quantitative nature of the

neuroanatomical reconstructions at synaptic resolution are remarkable. Complementing the reconstructions with computational modeling and activity measurements, the study proposes a likely circuit function for a specific inhibitory cell type during behavior.

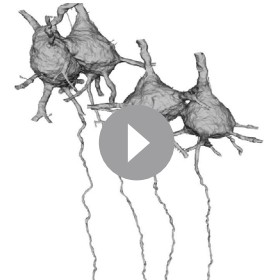

**Video 1.** A rendering of the electron microscopy (EM) reconstructions from this dataset demonstrating the mapping of chandelier cell (ChC) inputs onto layer 2/3 pyramidal neurons (PyCs). Video begins with four gray PyCs with only their somatic regions and axon initial segment (AIS) region shown. An individual pink ChC fragment is slowly revealed over time as the reconstruction is followed along to all the locations that it synapses onto. Note that the portions of that axon that are far from the four PyCs are excluded from the rendering for clarity. Then a second, purple axon fragment is revealed in the same fashion. Third, all the ChC fragments that synapse onto these four PyCs are revealed simultaneously, each with their unique color. Finally, the scene reveals all the PyCs in this structural dataset, and all the ChC branches reconstructed in the dataset are revealed in red.

https://elifesciences.org/articles/73783/figures#video1

## Introduction

The diversity of inhibitory cell types in mammalian neocortex, each with distinctive projection patterns and physiology, implies a rich role in cortical computation (*Petilla Interneuron Nomenclature Group et al., 2008*; *Fino et al., 2013*; *Freund and Buzsáki, 1996*; *Jiang et al., 2015*; *Kepecs and Fishell, 2014*; *Kubota, 2014*). To understand the role of a particular cell type in the brain, knowledge about the cell-type identity, connectivity within and between cell types and activity is required. For example, vasoactive intestinal polypeptide (VIP)-positive interneurons preferentially inhibit other inhibitory interneurons (*Lee et al., 2013*; *Pfeffer et al., 2013*; *Pi et al., 2013*) and are active during behaviors such as locomotion (*Fu et al., 2014*) and whisking (*Lee et al., 2013*; *Muñoz et al., 2017*). This established a key role for VIP neurons in the dynamic regulation of cortical inhibition under the control of brain-state-dependent modulators (*Alitto and Dan, 2012*; *Kawaguchi, 1997*; *Lee et al., 2013*; *McGinley et al., 2015*; *Muñoz et al., 2017*; *Pakan et al., 2016*; *Pi et al., 2013*; *Polack et al., 2013*; *Reimer et al., 2014*; *Stryker, 2014*; *Vinck et al., 2015*; *Zhang et al., 2014*). Similar efforts have elucidated circuit roles of other cell types, including parvalbumin (PV)-expressing basket cells (*Atallah et al., 2012*; *Nienborg et al., 2013*; *Packer and Yuste, 2011*; *Wilson et al., 2012*) and somatostatin (SST)-expressing Martinotti cells (*Adesnik et al., 2012*; *Muñoz et al., 2017*; *Nienborg et al., 2013*; *Silberberg and Markram, 2007*; *Wang et al., 2004*; *Wilson et al., 2012*).

The chandelier cell (ChC) has properties that, at first glance, should make it a good candidate to be among the better-understood inhibitory cell types. Sometimes referred to as axo-axonal cells, ChCs are GABAergic interneurons-characterized vertical axonal 'candles' or 'cartridges' (*Jones, 1975*; *Peters et al., 1982*; *Szentágothai and Arbib, 1974*) that synapse almost exclusively with the axon initial segment (AIS) of excitatory pyramidal neurons (PyCs) (*DeFelipe et al., 1985*; *Fairén and Valverde, 1980*; *Somogyi, 1977*; *Somogyi et al., 1982*). This pattern of connectivity suggests a unique role for ChCs as the AIS is a specialized compartment whose unique ion channel distribution makes it the principal site of action potential generation (*Kole and Stuart, 2012*; *Kole et al., 2007*; *Palmer and Stuart, 2006*). Each ChC connects to 30–50% of PyCs within a 200-µm-wide axonal field (*Wang et al., 2019*), resulting in a small number of ChCs innervating a large number of PyCs in L2/3 (*Inan et al., 2013*). It follows that ChCs are particularly well positioned to exert powerful control over PyCs in superficial layers and, as a result, the entire cortical activity.

Despite their highly specific connectivity, how ChCs affect neuronal circuits remains enigmatic (*Inan and Anderson, 2014*; *Woodruff et al., 2010*). One reason is the variability in the strength of ChC targeting. PyC populations in different brain regions and layers are contacted by diverse numbers of AIS-targeting boutons (*DeFelipe et al., 1985*; *Veres et al., 2014*; *Wang and Sun, 2012*). Consistent with this observation, activation of the ChC population in vivo has diverse effects on nearby PyCs, ranging from strong inhibition to a lack of response (*Lu et al., 2017*) despite apparently dense ChC connectivity (*Inan et al., 2013*). The logic underlying this variability, and thus the heterogeneity of ChC influence on the downstream targets, is largely unknown, although there are indications that

PyCs with different long-range projection targets can have different magnitudes of ChC inhibition (*Fariñas and DeFelipe, 1991b*; *Lu et al., 2017*). A second reason why the role of ChCs has remained enigmatic is functional. While several individual sources of synaptic input into ChCs have been identified (*Jiang et al., 2015*; *Lu et al., 2017*), what drives ChC activity and how that, in turn, manifests itself into circuit-wide dynamics in a behaving animal remain unknown.

To improve our understanding of both the circuit organization and function of ChCs, we took a multipronged approach. We used large-scale serial-section electron microscopy (EM) (*Bock et al., 2011*; *Kasthuri et al., 2015*; *Lee et al., 2016*) to map synaptic input onto the AIS across L2/3 PyCs in a volume of mouse primary visual cortex (*Video 1*). By using automated dense segmentation and synapse detection, we obtained a reconstruction of ChC axons and PyCs in a volume of approximately $3.6 \times 10^6$ µm$^3$. The resolution and completeness afforded by this approach allowed us to infer underlying principles governing not only the presence but also key properties of ChC connectivity. In particular, it allowed us to study the variability of the synaptic connectivity onto different pyramidal cells and compare it with perisomatic inhibition, which is formed by different cell types. To address how ChC inhibition influences PyC activity at the cellular level, we use biophysical simulations and map input-output relationships that show the unique impact of axo-axonic over other forms of inhibition. Finally, we use a novel genetic approach to selectively measure ChC activity in the visual cortex of awake behaving mice being presented with visual stimuli.

## Results

### A densely segmented EM volume of layer 2/3 primary visual cortex

EM offers the ability to identify every synapse and trace every neurite in a volume of tissue. In order to make precise measurements of the structure of cortical circuits, we prepared, sectioned, and imaged an EM volume of layer 2/3 (L2/3) from primary visual cortex (*Figure 1A*) of a P36 male mouse, spanning approximately 250 µm × 140 µm × 90 µm, with 40-nm-thick sections imaged at 3.58 × 3.58 nm/pixel with transmission EM (*Figure 1B*; see Materials and methods for details). Performing EM for such a large cortical volume is necessary in order to measure connectivity at a sufficient degree of precision. Specifically, we needed to be able to follow a multitude of individual neuronal processes for large distances throughout the volume and comprehensively identify synapses. Such circuit reconstruction at scale also requires intensive computational processing (*Berning et al., 2015*; *Dorkenwald et al., 2017*; *Jain et al., 2010*; *Januszewski et al., 2018*). We used a series of novel machine learning-based methods to perform image alignment, automated segmentation, and synapse detection for the volume (*Figure 1C*; *Turner et al., 2020*). Nonetheless, proofreading remains necessary for precise measurements of anatomy and connectivity. The initial segmentation identified small supervoxels that were agglomerated into cells, and we built a novel cloud-based proofreading system to edit the agglomerations and perform targeted error correction (*Dorkenwald et al., 2019*; *Dorkenwald et al., 2020a*) (see Materials and methods). The same dataset has also been used concurrently in other studies (*Buchanan et al., 2021*; *Dorkenwald et al., 2019*; *Turner et al., 2020*). We conclude that we established a high-throughput EM pipeline that, combined with novel image-processing computational tools, allows the ability to reconstruct entire cortical circuits at unprecedented spatial detail.

### A complete map of synaptic input to the AISs of an excitatory network

We manually identified all cell bodies in the volume (n = 547) and manually classified each as excitatory PyCs (n = 416), inhibitory (n = 34), or glia (n = 97) based on morphology and ultrastructural features such as dendritic spines. Dendritic and axonal arbors of all neurons with cell bodies were proofread to correct segmentation errors. The full dataset, including soma classifications, can be explored online at https://microns-explorer.org/chc/soma/all_by_type.

We next mapped all synaptic input onto the AIS of excitatory cells that had a complete AIS in the volume. Since we could not robustly identify the molecular components of the AIS from EM imagery, we opted instead for a purely structural definition: from the axon hillock (whether it emerged from the soma or a proximal dendrite) to the most proximal of the first branch point, beginning of myelination, or the volume exit. Only cells with at least 40 µm of AIS within the volume were considered, a distance found to contain almost all AIS synapses on more complete reconstructions (see also *Figure 2I*). For each PyC with a complete AIS (N = 153), we manually marked points as the top and bottom of the AIS

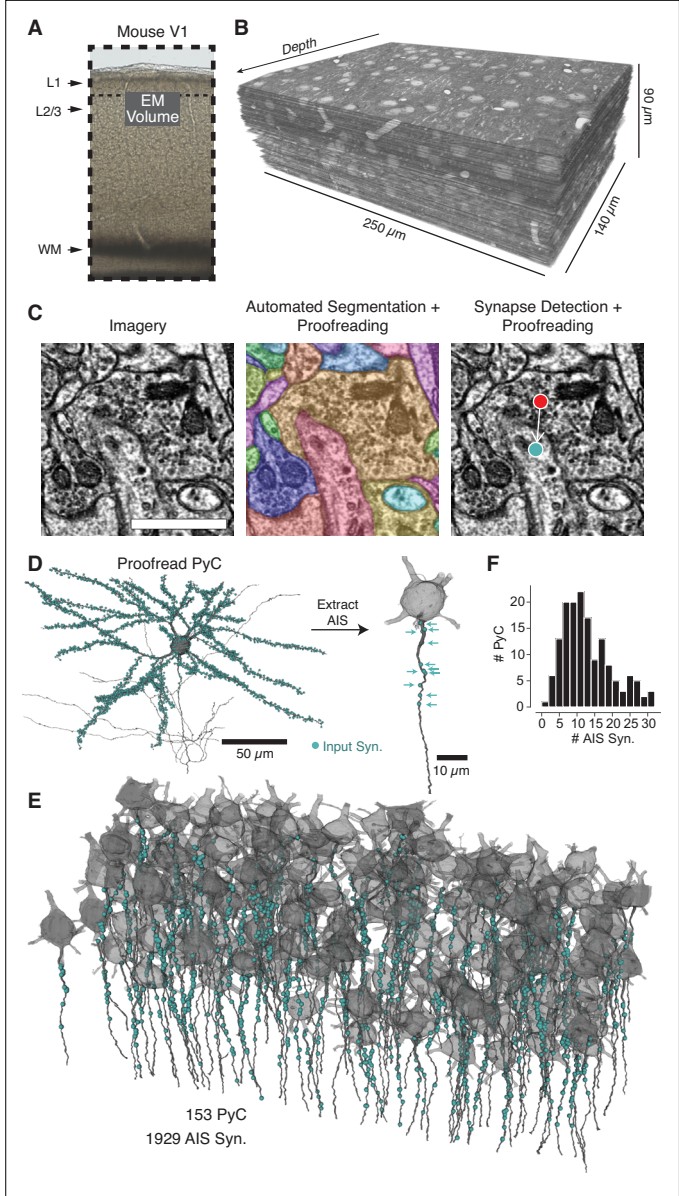

**Figure 1.** A map of axon initial segment (AIS) input from electron microscopy (EM). (**A**) A block of tissue was selected from L2/3 of mouse V1 and processed in an EM pipeline. (**B**) Serial 40 nm sections were imaged computationally aligned into a volume. (**C**) Image annotation pipeline. Left: images were taken with 3.58 × 3.58 nm pixels. Scale bar is 500 nm. Center: the neuropil was densely segmented and targeted proofreading was done to correct pyramidal neurons (PyCs) and other objects of interest. Right: automated synapse detection identified pre- and postsynaptic locations for synapses and was followed by targeted proofreading for false positives. To interactively view the dataset in 3D, visit https://www.microns-explorer.org/chc/soma/all_by_type. (**D**) For each PyC with significant AIS in the volume, we started with the overall morphology and synaptic inputs (cyan dots) and computationally extracted the AIS (dark gray) and its synaptic inputs (cyan arrows). All AIS bounds can be seen in 3D at https://www.microns-explorer.org/chc/soma/ais_bounds. (**E**) Soma, AIS, and AIS synaptic inputs for all PyC analyzed. The volume is rotated so that the average AIS direction is exactly downward. Note that dendrites and higher-order axon branches are omitted for clarity. (**F**) Histogram of synapses per AIS. Synapse data can be found in *Supplementary file 2*.

The online version of this article includes the following figure supplement(s) for figure 1:

**Figure supplement 1.** Axon initial segment (AIS) segmentation pipeline.

**Figure supplement 2.** Example synapse imagery for all axon initial segment (AIS) inputs onto cell ID 648518346349535847.

*Figure 1 continued on next page*

*Figure 1 continued*

**Figure supplement 3.** Example synapse imagery for all axon initial segment (AIS) inputs onto cell ID 648518346349539517.

**Figure supplement 4.** Example synapse imagery for all axon initial segment (AIS) inputs onto cell ID 648518346349539572.

and used them to computationally specify the AIS and its synaptic input (*Figure 1D and E*, *Figure 1—figure supplement 1*, https://microns-explorer.org/chc/soma/ais_bounds). All synapses were proofread to remove false positives from the automated synapse detection. This resulted in a total of 1929 AIS synapses across 153 PyCs, for a mean of 12.6 synapses per AIS (for examples, see *Figure 1—figure supplements 2–3* and *Supplementary file 1*). Importantly, we observed a remarkable diversity of total inputs, ranging from 1 to 32 synapses per AIS (*Figure 1F*), pointing to differing magnitudes of AIS innervation across the PyC population.

## PyC AIS input is a mix of chandelier and non-chandelier synapses

While ChCs are the only cell type to specifically target the AIS, other cell types can also form synapses on the AIS (*Gonchar et al., 2002*; *Gour et al., 2021*; *Kisvárday et al., 1985*; *Somogyi, 1977*). To identify which AIS input synapses came from ChCs and which did not (*Figure 2A*), we examined the morphology and ultrastructure of every axon presynaptic to the AIS of any of the PyCs (*Figure 1—figure supplements 2–3*, *Supplementary file 1*). By following all axons forming synapses onto AIS of the 153 PyCs considered in the cortical volume, we found axons that exclusively target the AIS of PyCs (*Figure 2B*, *Figure 2—figure supplement 1*) and axons that targeted a mixture of compartments (*Figure 2C*). Axons exclusively targeting AIS were manually proofread and re-evaluated, while mixed-target cells were proofread to discover any falsely merged AIS-targeting fragments. AIS-exclusive targeting axons were classified as ChCs, while mixed-target axons were classified as non-ChCs (see Materials and methods). In total, we found 1127 AIS synapses from ChCs (122 axon fragments and 2 ChCs with soma in the volume, see *Figure 2—figure supplement 2*) and 802 AIS synapses from non-ChC axon fragments (*Figure 2D and E*). Each AIS and its complete presynaptic input can be viewed online (e.g., https://www.microns-explorer.org/chc/ais/1 — see *Supplementary file 1* for all URLs). We conclude that while the majority of AIS synapses in our volume originate from ChC axons, more than 40% originate from non-ChC axons.

To understand how ChC and non-ChC axons differ in how they target the AIS, we looked at the precise location of their synapses. We found that ChC synapses onto PyCs were predominantly located on the AIS in a region between 10 and 40 µm from the axon hillock, consistent with previous observations (*Veres et al., 2014*), while non-ChC synapses were most common near the soma, but widely distributed across the AIS (*Figure 2F*). Indeed, ChC and non-ChC synapses were often intermingled within individual AISs (*Figure 2G–I*). Even within the limited span of the AIS, ChC axons were not distributed randomly. From the 3D reconstructions of ChC boutons targeting a single AIS, we noticed that instead of being scattered on the surface of the AIS (as observed for somatic inhibitory input), multiple ChC boutons from different axons were clustered at distinct points on the AIS (*Figure 2—figure supplement 3*). Under such ChC clusters, we typically observed a cisternal organelle (CO), an AIS-specific endoplasmic reticulum specialization associated with a complex assortment of molecular components (*King et al., 2014*) that actively contribute to the local modulation of calcium (*Lipkin et al., 2021*). Consistent with previous qualitative observations (*Benedeczky et al., 1994*; *King et al., 2014*), we found that ChC synapses were significantly closer to COs than expected by spatially shuffling synapse locations within and between AISs (*Figure 2—figure supplement 4*). Taken together, our results show that not only do ChCs target a more specific region on the AIS than non-ChCs, but that ChC boutons frequently form clustered synapses with other ChCs at the site of a specialized postsynaptic organelle.

## Distribution of AIS input by cell type

An AIS can receive input from many different ChCs, each with potentially multiple synaptic contacts. We use 'synapse' to refer to a single anatomical synapse and 'connection' to indicate the collection of synapses between a given presynaptic axon branch and postsynaptic AIS, comprising one or more synapses. The ability to discriminate each presynaptic axon in our data offers the opportunity to look

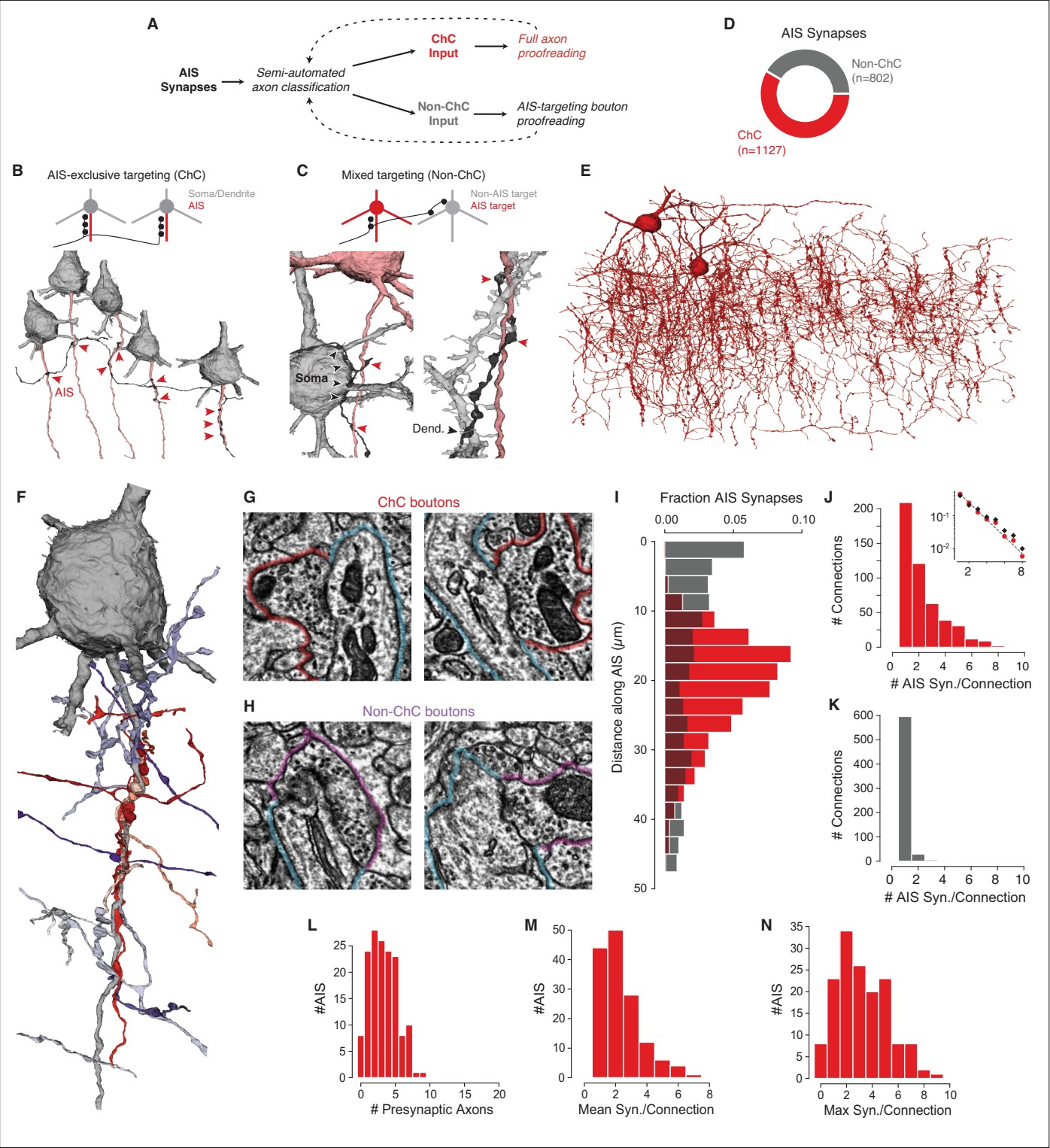

**Figure 2.** Characterization of axon initial segment (AIS) input. (**A**) Classification and proofreading workflow. Across AIS synaptic inputs, morphology and connectivity were used to distinguish chandelier cell (ChC) from non-ChC axons. ChCs were given full proofreading to get as-complete-as-possible arbors. For non-ChC inputs, all AIS-targeting boutons were proofread to ensure they were not due to merged-in ChC axons. The pipeline was repeated until no new axons were identified. (**B**) Axons that exclusively synapsed onto pyramidal neuron (PyC) AISes were classified as ChCs. Example below shows an axon targeting the AIS of several PyCs (red arrowheads). (**C**) Axons that showed mixed AIS and non-AIS targeting were classified as non-ChC. Example at left shows an axon (black) targeting an AIS (red arrowheads) and a nearby soma (black arrowheads). Example at right shows an axon (black) targeting a dendrite (black arrowheads) and an AIS (red arrowheads). (**E**) All ChC objects identified from the AIS survey. (**F**) Single PyC soma and AIS

*Figure 2 continued on next page*

*Figure 2 continued*

shown with presynaptic axons. Reds indicate ChCs, purples indicate non-ChCs. Axons are truncated to the region near AIS contacts for clarity. For the full data in 3D, please visit https://www.microns-explorer.org/chc/ais/80. (**G**) Example ChC synapse imagery for the AIS in (**F**). The AIS outlined in blue, ChC bouton in red. Image panels are 1 μm × 1 μm. (**H**) Example non-ChC synapse imagery onto the AIS in (**F**). As in (**G**) but with purple outlines for boutons. (**I**) Histogram of distance from AIS base for ChC (red) and non-ChC (gray) synapses. (**J**) Distribution of AIS synapses per connection for ChCs. Inset: semilog-y synapses per connection for all ChCs (red dots) and only ChC axon fragments with more than 20 synapses (black dots), with Poisson fit to the complete data (dashed line). (**K**) Distribution of AIS synapses per connection for non-ChCs. (**L**) Number of distinct presynaptic ChC axons per AIS. (**M**) Distribution of mean number of synapses in a ChC connection for each AIS. (**N**) Distribution of the number of synapses in the most numerous ChC connection on each AIS. Synapse data can be found in *Supplementary file 2*.

The online version of this article includes the following figure supplement(s) for figure 2:

**Figure supplement 1.** Characterization of orphan axons.

**Figure supplement 2.** Examples of chandelier cell (ChC) dendrites and dendritic input.

**Figure supplement 3.** Chandelier cell (ChC) input is located near cisternal organelles (CO) on the axon initial segment (AIS).

**Figure supplement 4.** Clusters of chandelier cell (ChC) inputs on the axon initial segment (AIS).

at the complete map of synaptic input and how it is organized by the identity of the presynaptic axon. It is likely that multiple axon branches belong to the same few ChCs (either the two found in the volume or others outside), but this identity cannot be reconstructed within the volume. However, based on whole-cell reconstructions showing that different branches of ChC axons rarely converge onto the same cartridge (*Blazquez-Llorca et al., 2015*; *Gouwens et al., 2019*), it is also likely that most distinct ChC axon branches targeting the same AIS come from distinct cells. However, such convergence cannot be ruled out.

Individual ChC connections had between 1 and 9 synapses, and their distribution was well-fit by a geometric distribution (exponent: 0.44, *Figure 2J*). Notably, the synapse count included both standard multisynapse connections that characterize ChC cartridges, as well as numerous single-synapse connections. This distribution of connections was observed even when we only considered larger axon fragments (more than 20 synapses) (*Figure 2J*, inset), suggesting that it is not due only to highly fragmented reconstructions. In contrast, non-ChC connections comprised only a single synapse in 94% of examples (*Figure 2K*) and all non-ChC connections had three or fewer synapses in total. Our data suggest that while all high synapse-count AIS connections originate from ChC axons, both ChC and non-ChC form numerous weaker, often single-synapse, connections.

For a given AIS, the impact of ChC input depends not on a single connection, but on the combination of synapses from potentially multiple presynaptic ChCs (*Inan et al., 2013*). We found that individual PyCs received input from between 0 and 9 distinct ChC axons (mean 3.3 ChC axons per AIS) and those AIS that received ChC input had a mean of 2.3 synapses per connection (*Figure 2L and M*) and almost 80% of AISes (122/153) still had at least one cartridge with two or more synapse (*Figure 2N*). Thus, despite the frequency of weak ChC connections, due to convergent input from many cells most PyCs nonetheless have at least one typical ChC cartridge.

## ChC connectivity onto pyramidal cells is highly variable and correlates with properties of target cells

We next asked how ChC connectivity was related to the individual properties of the L2/3 PyCs targeted. Because different PyCs had different amounts of axonal arbor in the volume, we restricted our analysis to a consistent initial region that would both cover the typical molecularly defined AIS and include as many individual cells as possible. Based on the distribution of ChC synapses (*Figure 2F*), we used the first 37 μm of structural AIS for all cells, a span that contained 97% of ChC synapses (*Figure 3A*) while omitting only one PyC due to an insufficient length of AIS within the volume.

ChC synapses were found on 95% of PyCs (144/152), but there was striking variability in the total number of ChC synapses (mean: 7.4 ± 5.3 synapses), with individual PyCs receiving between 0 and 25 ChC synapses (*Figure 3B and C*). Effectively, some L2/3 PyCs escape ChC input entirely while others are strongly innervated by ChCs. We asked if this variability was reflected in nearby perisomatic input, which for PyCs is thought to be almost entirely inhibitory (*Davis and Sterling, 1979*; *Fariñas and DeFelipe, 1991a*; *White and Rock, 1980*; *Wildenberg et al., 2021*). We confirmed this in our data; 81/81 randomly sampled somatic synapses on five PyCs were from an inhibitory axon, as were

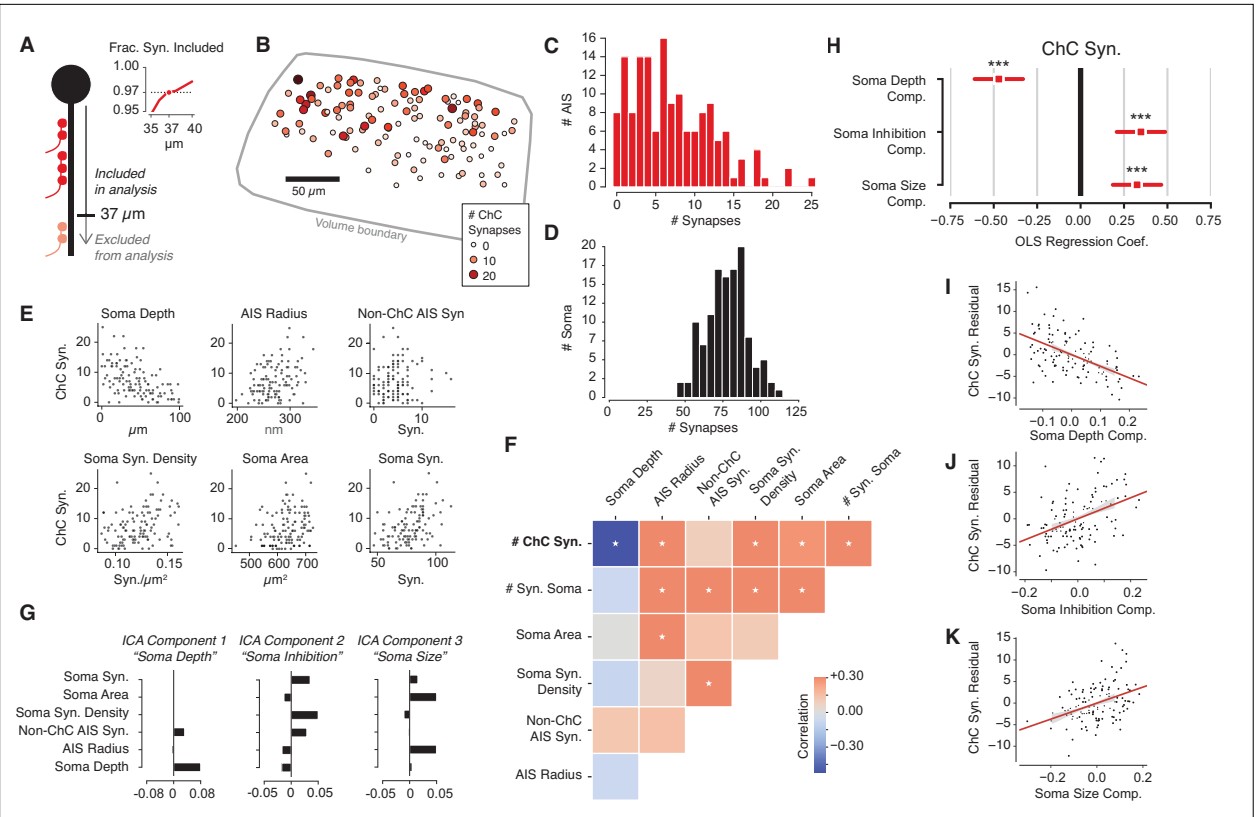

**Figure 3.** Structural properties associated with chandelier cell (ChC) synaptic input. (**A**) For each axon initial segment (AIS), we look only at synapses on the first 37 μm, which captures 97% of all ChC synapses while omitting as few cells as possible from analysis. (**B**) Soma location for cells with complete soma in the volume, colored by total ChC synapse count. Pia direction is up. (**C**) Distribution of total ChC synapse count across pyramidal neurons (PyCs). (**D**) Distribution of total somatic synapses across PyCs. (**E**) ChC synapse count vs. six structural properties of soma and AISes: depth, AIS radius, non-ChC AIS synapse count, soma synapse density, soma area, and soma synapse count. (**F**) Pearson correlation matrix between all structural properties. Entries with an asterisk are significant (p<0.05) after Holm–Sidak multiple test correction. (**G**) Independent components analysis (ICA) components for PyC structural properties. Components are oriented so that the highest loading element is positive. Each component is labeled with an approximate interpretation of its combination of properties. (**H**) Standardized ordinary least-squares (OLS) regression coefficients for ChC vs. the three ICA components. Bars indicate 95% confidence interval; stars indicate significance after Holm–Sidak multiple test correction. ***p<0.001. (**I**) Scatterplot and linear fit of residual ChC synapse count vs. depth after fitting the other two components. Shaded region indicates the 95% confidence interval estimated from bootstrap (N = 1000). (**J**) Same as (**I**), but for the soma inhibition component. (**K**) Same as (**I**), but for soma size component. Synapse and AIS data can be found in *Supplementary file 2*.

The online version of this article includes the following figure supplement(s) for figure 3:

**Figure supplement 1.** Soma synapse labeling and measurements.

**Figure supplement 2.** Distribution of synapses in the volume.

**Figure supplement 3.** Scatterplots of pyramidal neuron (PyC) structural independent components analysis (ICA) components.

all somatic synapses (N = 75) on an additional randomly selected PyC (see Materials and methods). Using similar methods as for AIS synapses, we identified all somatic synapses for the 120/152 PyC whose cell bodies were fully contained in the volume (*Figure 3—figure supplement 1*). All PyCs had numerous somatic input synapses (47–113 synapses, *Figure 3D*). Notably, among those cells with fully measured somatic synapses, the total number of ChC synapses had a coefficient of variation (CV) of 0.73, compared to a CV of 0.18 for the number of perisomatic PyC synapses. We conclude that axo-axonic ChC synapse counts exhibits substantially higher variability than perisomatic synapse counts for PyCs.

To explore the logic of the observed heterogeneity, we asked if the total number of ChC synapses that a PyC receives was associated with other structural properties of the target cell. For consistency in our measurements, we focused on the soma and the AIS. Specifically, for each PyC with a complete soma, we measured its depth within L2/3, mean AIS radius, number of non-ChC AIS synapses, number

of synaptic inputs onto the soma, soma surface area, and soma synapse density. Strikingly, we found statistically significant correlations between the number of ChC synapses and each property except for non-ChC AIS synapses (*Figure 3E and F*). However, we also found that various size and synaptic input properties of each PyC were significantly correlated among themselves (*Figure 3F*). To disentangle these correlations, we performed independent components analysis (ICA), a variant of principal components analysis that yields fully uncorrelated components (*Comon, 1994*). ICA attributed the observed correlations to three main postsynaptic PyC components: the cortical depth of the soma (i.e., location), the amount of somatic inhibition (i.e., soma synapses, soma synapse density, and non-ChC AIS synapses), and the perisomatic PyC size (i.e., soma area and AIS radius) (*Figure 3G*).

We next asked if these components were associated with differences in total number of ChC synapses. We performed multivariate ordinary least-squares (OLS) regression on the number of ChC synapses against the three ICA components for each PyC. All three components showed significant correlation with the number of ChC synapses, with the three together explaining 45% of the variance in ChC synapse number (*Figure 3H–K*). We report coefficients from z-scored variables. First, deeper PyCs received fewer ChC synapses (coefficient: –0.47). This is consistent with the observation that some ChC axons have denser axonal arbors in upper L2/3 (*Wang et al., 2019*) and our observation of a higher absolute number of ChC synapses in the upper part of our volume (*Figure 3—figure supplement 2*). Second, perisomatic inhibition was positively correlated with the number of ChC synapses (coefficient: 0.33) — that is, PyCs with more ChC synapses also receive more synapses from other, non-ChCs at their soma. Third, larger cells received more ChC synapses (coefficient: 0.34). Taken together, our data support that the amount of ChC synapses onto PyCs is influenced by the shape, location, and inhibitory connectivity of each target cell.

## Properties of individual target cells influence the number of chandelier synapses on the AIS

The same number of ChC synapses onto a single PyC could result from different combinations of the number of connections and the number of synapses per connection. (*Figure 4A*). Variability in the total number of ChC synapses along PyCs could thus be attributed to either or both connectivity properties. To understand the distinct role of both of these connectivity properties separately, we measured both the number of connections (i.e., unique presynaptic ChC axon branches) and the number of synapses per connection for all PyCs. We found that both the number of connections and mean synapses per connection ranged widely (1–9 connections and 1–7 synapses per connection), but were uncorrelated ($r = -0.08$, p<0.36, *Figure 4B*). Moreover, using the same perisomatic components OLS approach described above, we found no significant relationship between synapses per connection and soma depth or inhibition, although there was a modest positive correlation with soma size (*Figure 4C and D*). In contrast, we found that the number of connections was related to all three PyC properties in a similar pattern as total ChC synapses (*Figure 4E and F*). This suggests that the number of distinct ChC axon branches contacting a cell and the number of synapses per connection are regulated by different processes.

Variability in the number of connections could be due to axon geometry alone; for example, if there were more ChC axons per AIS in some parts of the volume than others. To test this, we used a simple model to account for spatial effects based on *Stepanyants et al., 2002*. We define 'potential connections' as those ChC-PyC pairs where a ChC axon and AIS come within a given distance, and 'connectivity fraction' as the fraction of potential connections that are synaptically connected (*Figure 4G*). The number of connections can be decomposed into the number of potential connections times the connectivity fraction. If input variability were determined only by local abundance of ChC axons, then we would expect potential connections but not connectivity fraction to be related to PyC structural properties.

Because we know the location of every ChC axon branch in the EM volume, we can compute the potential connections and connectivity fraction for each AIS (AISs too close to volume boundaries were omitted, see Materials and methods). These values depend on the distance threshold used to define a potential synapse. We chose a distance threshold of 7.5 µm that resulted in connectivity fractions spanning 0–1, the full range of possible values (*Figure 4F*), although similar results hold for thresholds between 5 and 10 µm (*Figure 4—figure supplement 1*). While both potential connections and connectivity fraction exhibit statistically significant correlation with PyC properties, a different

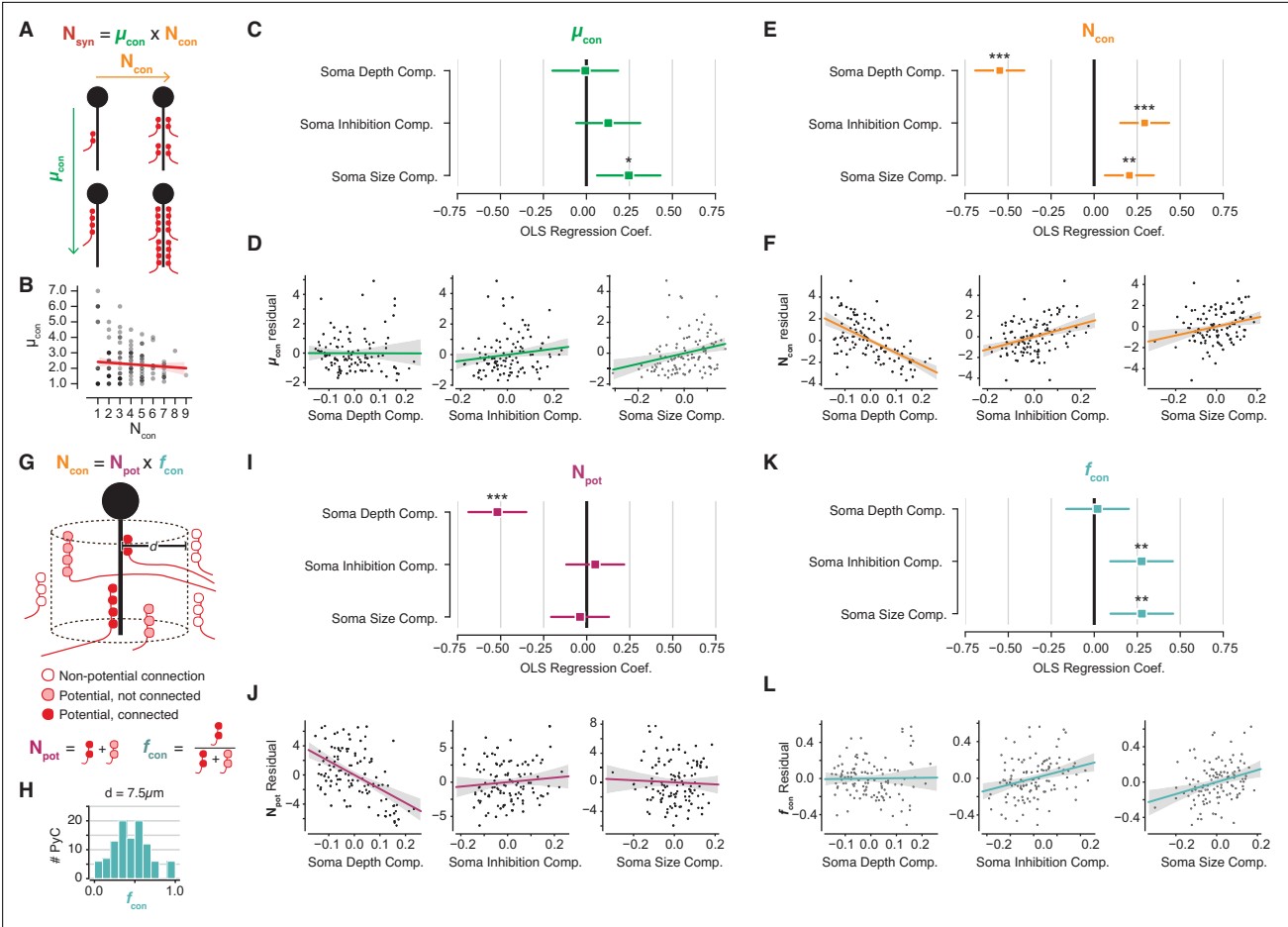

**Figure 4.** Decomposition of chandelier cell (ChC) synaptic input into constituent aspects. (**A**) Synapse count can be broken into mean synapses per connection ($\mu_{con}$) times number of connections ($N_{con}$). (**B**) Scatterplot of mean synapses per connection vs. number of connections for pyramidal neurons (PyCs) with soma in the volume. Red line indicates linear fit. (**C**) Standardized ordinary least-squares (OLS) coefficients for mean synapses per connection vs. independent components analysis (ICA) components, bars indicate 95% confidence interval. (**D**) Residual scatterplots of each ICA component vs. mean synapses per connection, as in *Figure 3H–J*. (**E**) Standardized OLS coefficients for number of connections. (**F**) Residual scatterplots for number of connections. (**G**) Number of connections can be broken into the number of potential connections ($N_{pot}$) near an axon initial segment (AIS) times the fraction of those potential connections that form actual synapses ($f_{con}$). (**H**) Distribution of $f_{con}$ for a potential radius of 7.5 µm, the value used for panels (**I–L**). Note that values span from 0 to 1. (**I**) Standardized OLS coefficients for potential connections. (**J**) Residual scatterplots for each ICA component vs. potential connections. (**K**) Standardized OLS coefficients for connectivity fraction. (**L**) Residual scatterplots for each ICA component for connectivity fraction. In all panels, stars indicate significance after Holm–Sidak multiple test correction. *p<0.05, **p<0.01, ***p<0.001.

The online version of this article includes the following figure supplement(s) for figure 4:

**Figure supplement 1.** Potential connectivity analysis for other potential distance radii.

set of properties dictates each correlation. On the one hand, the number of potential connections decreased with depth but had no significant relationship with soma size or soma inhibition (*Figure 4H and I*). In contrast, the connectivity fraction was not significantly correlated with soma depth, but was positively modulated by soma size and soma inhibition (*Figure 4J and K*). Together, these results indicate that while soma depth within L2/3 modulates the local abundance of ChC axons around a given AIS, cell size and inhibitory connectivity are correlated with the recruitment of ChC axons to an AIS.

## Biophysical simulations of chandelier inhibition on PyCs

Given the range and variability of ChC synapses observed along the AIS of PyCs, we wondered how such inhibitory innervation can influence the activity of PyCs. To explore how the number of synapses, number of connections, and the number of synapses-per-connection impact the spike output of PyCs, we used biophysical modeling. We used four biophysically detailed, single-cell models of cortical V1

excitatory neurons generated by an automated, multiobjective genetic optimization workflow. The starting point of the workflow are slice electrophysiology experiments (whole-cell patch-clamp) and the reconstructed morphologies of the same neurons (*Nandi et al., 2020*; see Materials and methods) while the single-cell models contain a number of active ionic (Na, K, and Ca) conductances along their morphology, including the AIS. The resulting morphologically accurate conductance-based PyC models reproduce a number of physiological properties (*Figure 5A–C*). Importantly, these PyC models are ideally suited to study the effect of excitatory and inhibitory inputs impinging along the morphology as one has access to all variables and parameters of a model during simulations.

We used modeling to ask how the location of inhibition along the PyC morphology impacts the output of pyramidal neurons in the presence of excitatory inputs along the dendritic arbor. We were particularly interested in the effect of AIS-targeting inhibition (mimicking ChCs) and thus considered four scenarios: one with excitation only and three with different temporal patterns of inhibition that we term *synchronous*, *biologically inspired,* and *asynchronous* (*Figure 5D*). First, in the synchronous inhibition case, all ChC synapses are activated by the same realization of a Poisson process, resulting in inhibitory input synchrony. Second, in the biologically inspired inhibition case, clusters of ChC synapses are activated synchronously, but each cluster through a distinct realization of a Poisson process, simulating the activity of multiple presynaptic ChC (each with one or more synapses) converging onto the same AIS. In this case, the AIS inhibition is spatiotemporally heterogeneous. The number of synapses per cluster (i.e., per presynaptic axon) was drawn from measured PyCs in the EM data, chosen to span the range of total input synapse counts and assumed each ChC axon branch came from a distinct cell. Finally, in the asynchronous inhibition case, each ChC synapse is activated by an independent realization of the Poisson process, resulting in co-active ChC input with uncorrelated spike times. As expected, in the presence of excitatory synaptic input onto PyC dendrites, ChC inhibition reduced spiking output of the PyC with the three scenarios exhibiting differences (*Figure 5D and E*). Specifically, the asynchronous ChC input was most effective at suppressing PyC output, while the synchronous ChC input was least effective. The biologically inspired scenario presented an intermediate case between the synchronous and asynchronous ChC input in terms of suppressing PyC output. These trends remain robust when increasing the excitatory drive as well as when altering the number of ChC synapses along the AIS across PyC models (*Figure 5E and F*). We conclude that the AIS innervation by multiple presynaptic ChCs resulting in spatiotemporally heterogeneous inhibition is stronger in terms of suppressing PyC output than if all the synapses were provided by a single ChC.

How does the innervation location of ChC inhibition affect PyC output or, alternatively, what is the effect of ChC specifically targeting the PyC AIS over other compartments? We looked into the effect exerted by ChC inhibition when targeting the PyC AIS vs. the soma. Using the same four excitatory PyC models, we simulated Gaussian barrages of dendritic excitation and either AIS or somatic inhibition with different temporal offsets, from inhibition leading excitation by 100 ms to inhibition trailing excitation by 100 ms (*Figure 5G*). In general, temporal disparity between the excitatory and inhibitory barrage leads to overall higher PyC spike output compared to when the two barrages coincide in time. Yet, when the two barrages are coincident, AIS inhibition results in stronger suppression of PyC spiking compared to somatic inhibition (*Figure 5G and H*).

To make the effect of AIS vs. perisomatic inhibition more visible, we developed a metric we term 'differential effect ratio.' The differential effect ratio is calculated by the difference between AIS and soma spike counts divided by the sum of the spike counts (*Figure 5G and H*). The ratio ranges between –1 and 1, where 1 represents inhibition at the AIS being 100% more effective than at the soma. A ratio of 0, on the other hand, means no differential effect. We found that while for small and large numbers of ChC synapses the location of inhibitory synapses did not change PyC output, a statistically significant difference between AIS and somatic inhibition is observed for concurrent barrages when the number of ChC synapses along the AIS is between 4 and 15 (*Figure 5H*). We note that this intermediate range of ChC synapse number per PyC located along the AIS where the most prominent difference between axonal and perisomatic inhibition is observed coincides with the range observed experimentally (see *Figure 3*). We conclude that when the incoming excitation and inhibition barrages are co-active, the inhibition directly into the AIS exerted via ChC is significantly more effective in suppressing PyC output than somatic inhibition.

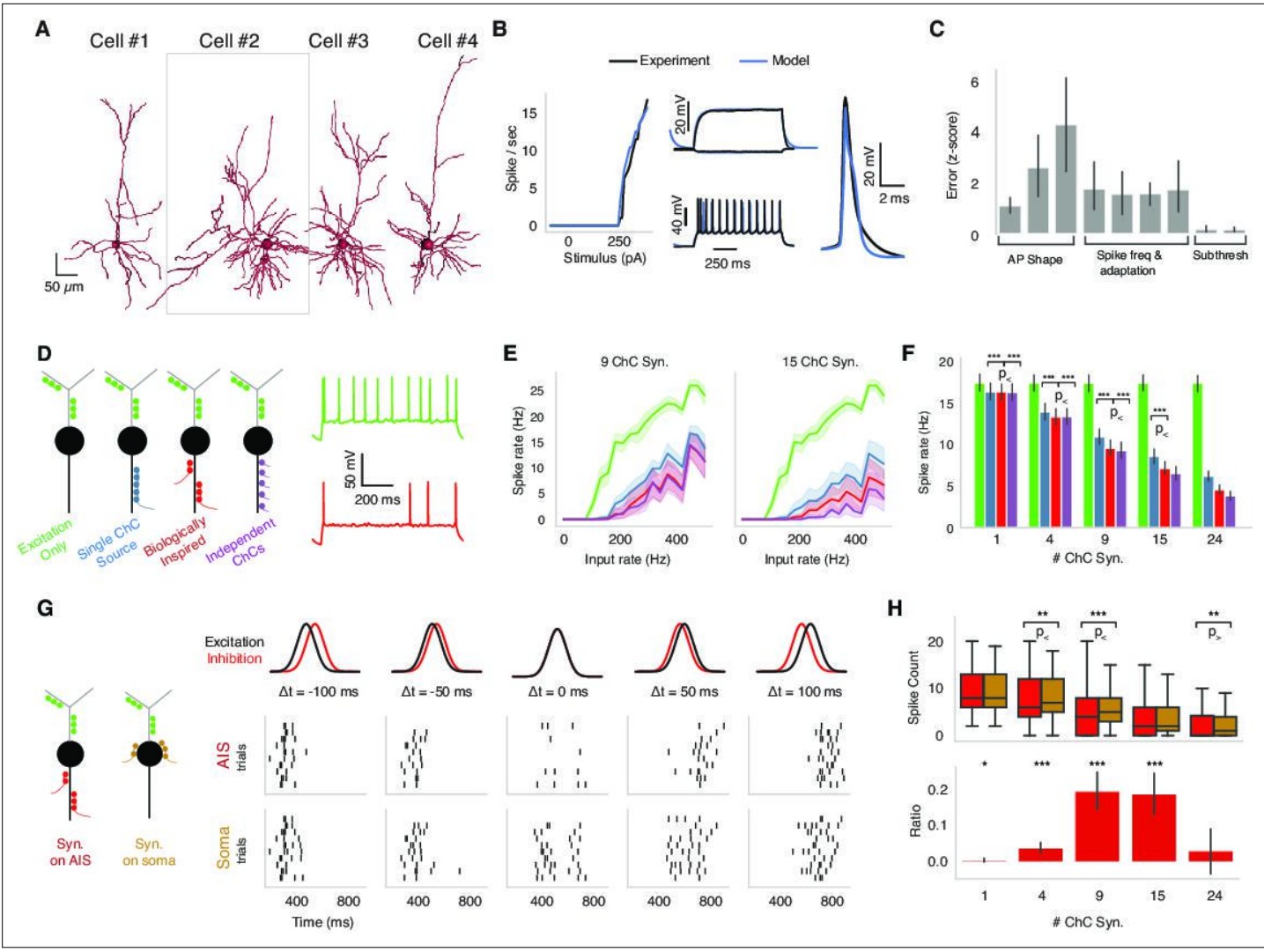

**Figure 5.** Biophysical modeling of chandelier cell (ChC) inhibition. (**A**) Reconstructed morphologies for four pyramidal neurons (PyCs) from mouse V1. (**B**) Biophysically detailed, ion-conductance models were fit for each cell constrained by its slice electrophysiology features and reconstructed morphology. Panels show model response vs. experimental data for one case (cell #2 in panel **A**). The panel shows the model performance in terms of *f-l*-response, spike waveform similarity and example subthreshold, suprathreshold somatic voltage traces (black) vs. the actual experimental traces (blue). (**C**) Average z-scored training errors for the four models (cell #1–4) for a set of electrophysiology parameters. (**D**) Left: schematic for different ChC configurations (green: dendritic excitation; colors along axon initial segment [AIS]: ChC inhibition innervation). Three inhibitory innervation patterns are considered that affect the temporal aspect of ChC synapse activation. Right: intracellular voltage response at soma when only excitation is active (green) vs. when AIS inhibition is co-active (red). (**E**) Increasing excitatory synaptic drive vs. PyC spike rate for the different inhibition scenarios (colors) for 9 vs. 15 total ChC synapses along the AIS. Results from one cell model (cell #2 in panel **A**) for different innervation realizations (line: mean; shaded area: SD). (**F**) Average PyC firing rate across the four single-cell models at fixed excitation across ChC configurations for increasing number of ChC synapses (bar: mean; error bar: SD; significance testing: Wilcoxon signed-rank test at 5% false discovery rate). (**G**) Effect of AIS vs. somatic inhibition barrages on PyC spike output (black: excitatory input barrage; red: inhibitory AIS barrage; left-to-right: difference between centroids is −100, −50, 0, 50, and 100 ms, respectively). Raster plots: spike output for AIS (top) vs. somatic (bottom) inhibitory barrage (all other properties remain identical). When the two barrages are coincident, AIS inhibition results in a decrease in PyC spiking vs. somatic inhibition. Data shown for multiple realizations in one model (cell #2 in panel **A**). (**H**, top) PyC spike output count for inhibition at the AIS vs. at the soma across four models and multiple realizations. (Bottom) The differential effect in inhibition as computed by the difference between soma and AIS spike counts divided by the sum of the spike counts. The ratio ranges between −1 and 1, where 1 represents inhibition at the AIS being 100% more effective than at the soma and 0 meaning no differential effect. The most prominent difference between AIS and somatic inhibition is for concurrent barrages when the number of ChC synapses along the AIS is between 4 and 15. Significance testing: Wilcoxon signed-rank test at 5% false discovery rate.

## Chandelier connections show no evidence of target selectivity beyond spatial proximity

So far, we have implicitly considered all ChC axons to be part of a uniform population. However, groups of ChC axon fragments could be clustered together, showing a tendency to co-innervate the same set of PyC AISs. If so, it would suggest that ChCs form functional subnetworks and are not, in

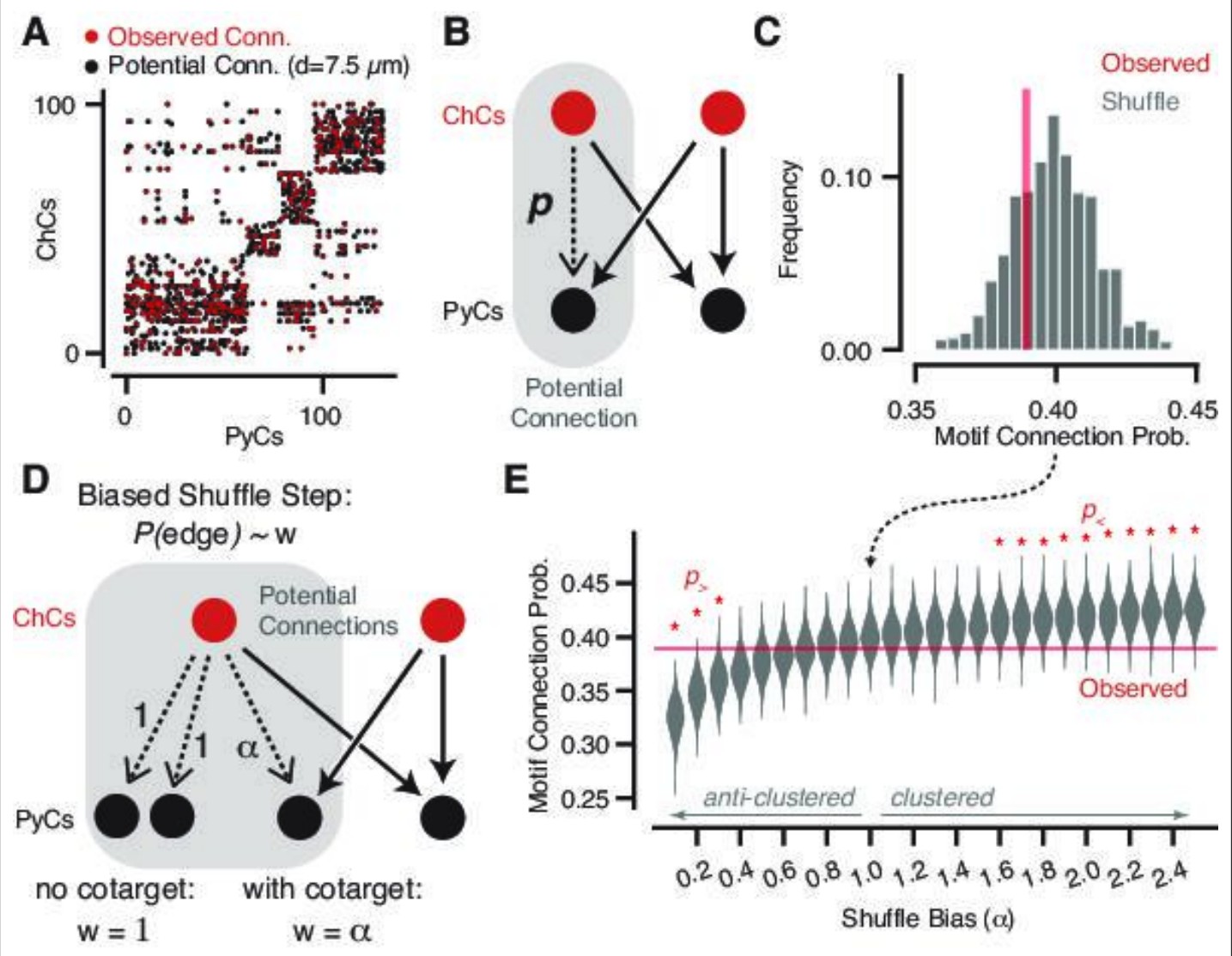

**Figure 6.** Geometrically constrained co-targeting motif analysis. (**A**) Connectivity matrix from chandelier cell (ChC) axons onto pyramidal neuron (PyC) targets. Each black dot represents one potential connection for a distance threshold of 7.5 µm, and each red dot represents one actual connection with any number of synapses. Elements are clustered by the potential connectivity matrix, suggesting that most of the structure of the network comes from geometry alone. (**B**) Cartoon of a potential bifan motif. In a bifan motif, two ChCs would target the same pair of PyCs. In a potential bifan motif, three of those connections are present and the fourth is a potential connection. We consider the probability p that this potential edge is connected. (**C**) Observed connectivity probability for the potential bifan motif in our dataset (red line) compared to shuffled networks that preserve PyC in-degree and potential connectivity. We see no evidence of excess clustering of ChC targets beyond geometry. (**D**) Cartoon of a biased bifan shuffle, where potential connections that complete a bifan motif are given a different weight in the shuffle probability. Setting the bias weight α > 1 encourages co-targeting by ChCs, while setting the bias weight α < 1 reduces co-targeting. (**E**) Observed connectivity (red line) probability vs. shuffled networks with different bias weights based on the shuffle step shown in (**D**) (gray violin plots, N = 1000 per bias weight value). Red stars indicate bias weights where the observed value is above (p>) or below (p>) 95% of the shuffled distribution. Even with the geometric constraints, we can rule out strong clustering or anticlustering within the data.

The online version of this article includes the following figure supplement(s) for figure 6:

**Figure supplement 1.** Geometrically constrained co-targeting motif analysis for other potential distance radii.

fact, a uniform population. When considering this question, we must also account for the physical distance between PyCs, otherwise proximity alone can introduce trivial correlations in their pattern of connectivity with ChC axons (*Figure 6A*). To account for both network clustering and distance, we measured the connectivity fraction of potential edges from ChCs to PyCs only for those that could complete a bifan motif when two ChCs target the same two PyCs (*Figure 6B*). If ChC connections were clustered, this motif-specific connection probability would be higher than that observed in networks where ChC connections were randomly shuffled amongst all the potentially connected ChC axons. The observed motif-specific connection probability we measured was well within the range obtained by shuffling (*Figure 6C*). We thus find no evidence that ChC axons are coordinated beyond geometrical factors that constrain potential connectivity.

To further control for this result and understand how strong a propensity for co-targeting we could detect given our dataset, we performed a power analysis by adding a motif-dependent bias to the shuffle. During the shuffle step, potential edges that could complete a bifan motif were selected with a weight $\alpha$, while those that did not were selected with weight 1. Thus, $\alpha > 1$ produces networks with increased clustering, $\alpha < 1$ biases against clustering, and $\alpha = 1$ is random (as above). The power analysis shows that we can detect deviations from random for a bias of $\alpha < 0.4$ or $\alpha > 1.5$, ruling out differential connectivity probability beyond a factor of 1.5–2.5. Therefore, the connectivity of chandelier axons onto pyramidal cells emulates that of random networks.

## In vivo function of ChCs shows collective activity during periods of arousal

The functional effect of ChC inhibition depends not only on the structure of synaptic connections, but also on the temporal pattern of activity, as observed in our modeling. At the circuit level, if all ChCs were active at the same time, total synapse count onto a PyC offers a good approximation of their net functional strength (*Veres et al., 2014*). On the other hand, activation of different ChC subnetworks at different times would suggest a more intricate relationship between structural connectivity and the influence of ChC activity on a given PyC.

To study how ChCs are activated in the cortical circuit during behavior, we used a mouse line in which recombinase CRE was coexpressed with Vipr2 (Vipr2-IRES2-Cre), a genetic marker expressed specifically in ChCs (*Daigle et al., 2018*). Full transgenic strategies do not label ChCs in mouse primary visual cortex effectively, likely due to off-target expression during development (*Tasic et al., 2018*). To circumvent this shortcoming, we injected an AAV viral vector containing a CRE-dependent calcium indicator gene GCaMP6f (*Chen et al., 2013*) into V1 of adult Vipr2-IRES2-Cre mice. Histological examination showed this strategy specifically labeled neurons at the layer 1/layer 2 border with arbors characteristic of ChC morphology (*Figure 7A*).

Using this strategy, we measured the in vivo calcium activity of ChCs by two-photon imaging. To simultaneously acquire imagery from sparsely distributed cells (*Figure 7B*), we used a multiplane imaging system (*Liu et al., 2018*) that allowed near-simultaneous monitoring across a range of cortical depth (*Figure 6C*). The imaging procedure followed a standardized awake behaving paradigm (*de Vries et al., 2020*), during which head-fixed mice were presented with a screen with uniform luminance and allowed to engage in spontaneous behavior.

In all imaging sessions, we observed striking seconds-long bouts where all ChCs were active, even with uniform luminance as a stimulus (*Figure 7D*). To quantify this, we calculated the average cell-cell correlation between ChCs during spontaneous behavior (*Figure 7E*, correlation coefficient, ChC-ChC: 0.49 ± 0.30, 68 pairs). The correlation among ChCs in the absence of dynamic visual stimulus was notable, even compared to other common interneuron cell types (correlation coefficient, PV-PV: 0.34 ± 0.24, 4,640 pairs; VIP-VIP: 0.18 ± 0.16, 5,717 pairs; SST-SST: 0.07 ± 0.20, 2,837 pairs), as estimated from the Allen Brain Observatory data (*de Vries et al., 2020*; *Figure 7E*).

Importantly, after testing the correlation between ChC activity for various behavioral states, we observed a robust relationship between coordinated ChC activity and pupil dilation: ChC activity occurred during periods of dilated pupils during locomotion, indicative of an active arousal state (*McGinley et al., 2015*; *Reimer et al., 2016*; *Vinck et al., 2015*). Notably, some pupil dilation events in the absence of locomotion had concurrent ChC activation (*Figure 7E*, gray bar). Indeed, ChC activity was more strongly correlated with pupil diameter than with locomotion speed (*Figure 7G*, correlation coefficient, ChC-pupil vs. ChC-locomotion: 0.40 ± 0.14 vs. 0.16 ± 0.11, 34 pairs, T =

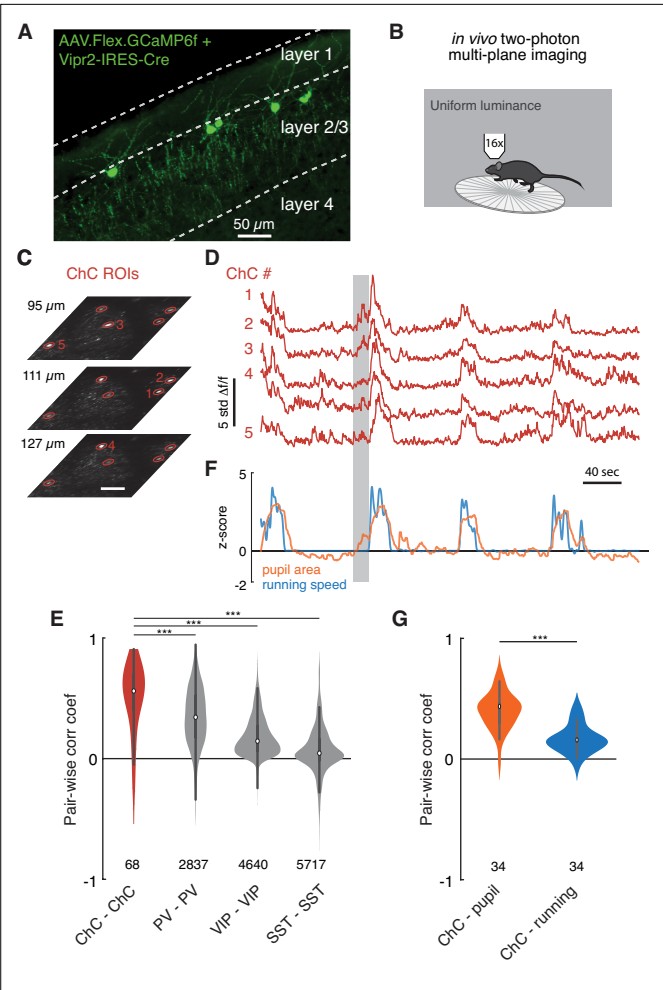

**Figure 7.** Functional imaging of chandelier cells (ChCs) reveals a synchronous response to arousal state. (**A**) Maximum projection image in upper layers of V1 showing ChC-specific GCaMP6f expression in Vipr2-IRES2-Cre mice injected with AAV-Flex-GCaMP6f. Note cell bodies at the L1/L2 border, characteristic cartridges, and L1 dendrites. (**B**) Cartoon of experimental design. Mice expressing GCaMP6f in V1 ChCs were placed on a treadmill and imaged with multiplane two-photon microscopy while subject to a uniform luminance visual stimulus. (**C**) An experiment with simultaneous three-plane imaging (depth shown on left) with five distinct ChC regions of interest (ROIs), each measured from its best plane. Scale bar is 50 µm. (**D**) GCaMP6f responses (z-scored from baseline) for the same five ROIs as in (**C**). Note extremely correlated large events. (**E**) Simultaneous behavioral measurements for the same experiment as in (**D**). Increased pupil size, a measure of arousal state, and running bouts correspond to periods of high ChC activity. Note that there are periods where ChC activity and pupil area increase in the absence of running (e.g., the period noted by the gray box). (**F**) Pairwise Pearson correlation for spontaneous activity traces for different interneuron classes. ChCs measured from two imaging sessions each for three mice show high spontaneous correlation compared to other classes of interneurons as computed from comparable observations in the public Allen Institute Brain Observatory data. Number of distinct pairs is shown. (**G**) Pairwise correlations between ChC activity traces and behavioral traces show that ChCs are significantly more correlated with pupil area than with running. Number of distinct pairs is shown. Statistical significance for cell-cell correlations was computed with the Mann–Whitney U test, while for matched cell-behavior correlations significance was computed with the Wilcoxon signed-rank test. ***p<0.001.

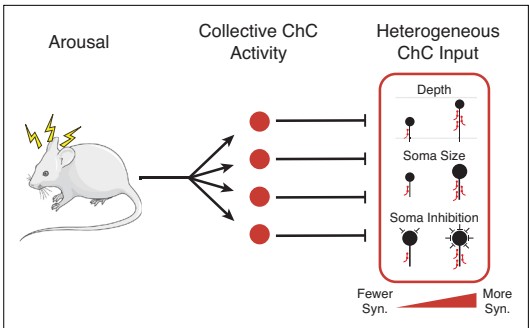

**Figure 8.** Cartoon summary of main results. Arousal drives collective activation of chandelier cells (ChCs) in visual cortex. This common signal is passed into pyramidal neurons (PyCs) in a heterogeneous manner, with input onto each target cell modulated by cell-specific structural properties.

3.0, p=4.8 × 10⁻⁷, Wilcoxon signed-rank test). These observations suggest that there are naturally occurring periods of high arousal when ChCs are collectively active. During these periods, L2/3 PyCs are likely to receive functional input from all or most of their presynaptic ChCs.

## Discussion

Using comprehensive measurements from volumetric EM data, we measured the complete presynaptic input onto the AIS of a large population of PyCs in mouse primary visual cortex and proofread all ChC axons connecting to them. We found that total ChC input per PyC was highly variable, but correlated to the depth, soma size, and net perisomatic inhibition of the target cells. Dissecting the role of individual axons, we found that depth was associated with ChC axon density, while size and inhibition effects were related to ChC connection probability. However, we could not find evidence of correlated patterns of input from specific ChC axons at the network level. We also used novel genetic tools to show that, during spontaneous behavior, ChCs are collectively active during periods of high arousal. Taken together, our results are consistent with a model where ChCs are synchronously activated during periods of arousal, providing a global signal to target neurons, but with the strength of that signal adjusted for each individual target neuron according to cell-specific structural factors (*Figure 8*).

### Comparison with previous measures of ChC connectivity

While the general features of ChC connectivity we observed are consistent with previous studies (*Pan-Vazquez et al., 2020*; *Tai et al., 2019*; *Veres et al., 2014*), we found strikingly more single-synapse ChC connections and non-ChC AIS inputs than have been previously appreciated. Undercounting single-synapse connections could happen for multiple reasons. Without a marker for ChC axons, it is difficult to distinguish in smaller EM volumes which GABAergic synapses on an AIS are from ChCs and which are not. Here, we solved this by using a large EM volume and tracing presynaptic axons far enough to use connection specificity to comprehensively identify non-ChC axons. Second, single-synapse connections are more ambiguous than multisynaptic cartridges, both from light-level anatomical images (*Inan et al., 2013*) and from electrophysiological recordings (*Veres et al., 2014*), and thus likely to be undercounted with such methods. Indeed, single *en passant* boutons consistent with the connections we found here are present in classic camera lucida illustrations (*Peters et al., 1982*) and more recent light-level study *Inan et al., 2013*; however, their contribution to synaptic connectivity has been unclear. The fragmentation of ChC axons in our data could also overestimate single-synapse connections if they target the same AIS as other axonal branches from the same neuron. While single-cell anatomical tracings show that such multiple contacts from the same axon are rare, it is possible that single-synapse connections are overrepresented in this scenario. Finally, at P36 the animal studied here is an adolescent (see discussion below), and while many properties of ChC axons have been observed to be consistent between P30 and P90, the morphology of ChCs continues to mature (*Inan et al., 2013*) and single-synapse connections could develop further or disappear subsequently. However, single-synapse connections could offer benefits in circuit function. Given the potential for activity-dependent plasticity in AIS input (*Pan-Vazquez et al., 2020*), weak connections could also allow ChCs to widely sample network activity without significantly altering it.

### Structured variability and implications for function

We found that ChCs formed synapses with nearly all PyCs in L2/3, in line with observations from mouse somatosensory cortex (*Inan et al., 2013*). However, the degree to which PyC received input from ChC axons exhibited striking heterogeneity even within a small volume. This heterogeneity in

the number of chandelier synapses onto the AIS is higher than the one seen in the synapses formed by basket cells in the perisomatic region. The wide distribution of chandelier synapses was already noted in early studies from cat and monkey (*DeFelipe et al., 1985*; *Fairén and Valverde, 1980*), but has remained largely unexamined. We found that three structural properties relating only to a PyC perisomatic region could explain approximately half the variance in its total number of synaptic inputs from ChCs: depth of the PyC within L2/3, the physical size of the cell, and the amount of somatic inhibition. Moreover, while the location of the PyCs affected the density of nearby ChC axons, size and inhibition were related to the likelihood of connection to nearby ChC axons. How might each of these properties reflect the function of ChCs on cortical circuits?

We found that deeper L2/3 PyCs received fewer ChC synapses than those nearer the L1/L2 border. The variation in inhibition across the depth of L2/3 (see also *Karimi et al., 2020*) suggests differences in the cellular or network properties of shallow L2/3 and deeper L2/3 PyCs. One possibility is differences in long-range targeting. In prelimbic cortex, ChCs are more likely to connect to shallow, amygdala-projecting PyCs than deeper, cortical-projecting PyCs (*Lu et al., 2017*). In mouse V1, intralaminar differences in the projections of L2/3 PyCs to higher-order visual areas have also been observed (*Kim et al., 2018*). Depth-dependent variability in ChC inhibition in V1 is thus well-posed to allow differential state-dependent inhibition of distinct excitatory subnetworks, although more work is necessary to show this relationship directly.

An increased number of ChC synapses was also associated with the physical size of PyCs soma and AIS. While this could arise from subtle differences in cell type (e.g., a subclass of PyC happens to be both larger and more strongly targeted by ChCs), an additional possibility is homeostatic balance (*Turrigiano, 2012*). All else being equal, the same connection onto a larger cell will result in a weaker postsynaptic response. A higher synapse count on larger cells could thus be a compensatory mechanism to achieve the similar responses across diverse cells. Such a homeostatic target point for ChC inhibition during postnatal development has been suggested by experiments that manipulated PyC activity and observed compensatory changes in the number of ChC boutons (*Pan-Vazquez et al., 2020*) as well as the location and size of the AIS (*Wefelmeyer et al., 2015*).

The positive correlation we observed between somatic input and ChC input also is consistent with a cell-specific target point for inhibition. As axons targeting soma are mainly from inhibitory cells, but not from ChCs, our data thus suggest that different inhibitory cell types are responsive to similar or correlated signals. Since different interneurons are functionally active at different times and under different conditions, this could be an effective way to ensure that the appropriate amount of inhibition is available across functional network states on a cell-by-cell level (*Vogels and Abbott, 2009*; *Vogels et al., 2011*). Interestingly, in our data, both somatic size and inhibition are associated with increased connection probability with nearby ChC axons, raising the possibility that both structural factors might engage similar mechanisms to form or prune connections.

One limitation of the dataset is that all cells and neuronal processes were truncated by the boundaries of the imaged volume. We took care to account for this truncation throughout our analysis by only studying neurons and compartments that we could map completely. However, this could introduce some biases. First, we have sampled the deeper part of L2/3 less than the shallow part since those axons left the volume closer to the soma. Deeper L2/3 PyCs than those we measured could have different properties. For example, if it is possible the decrease in ChC input with depth is a categorical step rather than a linear decline and we primarily sampled the transition region here. Second, most ChC axon branches exit the volume, and we cannot tell which of these are part of the same axonal arbors. The number of distinct axon branch connections is thus an upper bound on the number of distinct presynaptic ChCs, although light-level morphology suggests that it is rare for multiple ChC axonal branches to target the same AIS. It is thus possible that not all conclusions made at the level of axon branches will translate directly to input at the level of individual ChCs.

Future work in a larger dataset (*Bae et al., 2021*) could help not only resolve volume truncation issues, but also allow the use of richer data about both whole-cell morphology and a more complete synaptic network. Moreover, given that simple perisomatic features already accounted for approximately half the variance in ChC synapse count, it is possible that additional information could reveal additional factors related to ChC input, for example, functional activity, PyC subtypes, or excitatory network structure.

## Development state of chandelier connectivity

The EM data presented here describe the fine structure of axo-axonic connectivity at P36. Though this time point is immediately after the critical period (*Espinosa and Stryker, 2012*), it is possible that it does not reflect the final state of maturation as mouse adulthood is usually assumed to only be reached at P56. Consistent with the possibility that this cortex is still subject to plasticity, one can occasionally find locations where the ChC cartridge shows filopodia, suggesting that in those locations the connectivity is not fixed (see https://www.microns-explorer.org/chc/axon/filopodia). Whether these locations are a sign of local plasticity that remains present in the adult, or a sign that in these locations the circuit is still maturing will be something that can be tested in subsequent datasets in older animals (*Bae et al., 2021*) and at different status of development (*Gour et al., 2021*). However, previous work has shown little development variation of ChC cartridges between P18 and P90 (*Inan et al., 2013*), including the percentage of ChC cartridges contacting the AIS as well as number of boutons per cartridge, although continued lateral expansion of ChC axonal arbors was observed beyond P30. Perisomatic innervation, another parameter analyzed in this study, also appears to have reached a plateau in the fifth postnatal week (*Chattopadhyaya et al., 2004*), which is around the time of the study presented here.

## Functional role of ChCs

The functional role of ChCs has been far less studied than their structure, and it is not clear what conditions drive ChC activity. In both neocortex and in the hippocampus, ChCs have been shown to fire in a brain-state-dependent manner in anesthetized animals (*Klausberger et al., 2003*; *Massi et al., 2012*); however, their function in awake animals has not been previously described. Here, we used functional imaging of ChCs in awake behaving animals and observed strong, synchronous bouts of activity during spontaneous periods of pupil dilation, a key sign of an active arousal state (*McGinley et al., 2015*; *Reimer et al., 2016*; *Vinck et al., 2015*). Interestingly, during periods of pupil dilation ChCs were found to be tonically active, consistent with the activation patterns of layer 1 cholinergic axon projections (*Reimer et al., 2016*), which are known to innervate ChC dendrites (*Lu et al., 2017*).

Activity in visual cortex changes significantly during arousal, with PyCs generally reducing their spontaneous activity and increasing their signal-to-noise ratio (*Polack et al., 2013*; *Vinck et al., 2015*) and all major interneuronal classes in superficial layers showing increased gain in response to visual stimulus (*Pakan et al., 2016*). A subclass of VIP interneurons that are active during locomotion have been strongly implicated in this dynamic modulation of circuit inhibition, independent of visual stimulus (*Fu et al., 2014*; *Pakan et al., 2016*). Our data suggest that ChCs participate in the high-arousal inhibitory state, assuming ChCs in mature animals have an inhibitory function (*Rinetti-Vargas et al., 2017*). This suggests that ChCs add a significant extra source of inhibition to some – but importantly not all – L2/3 PyCs during the high arousal state. L2/3 PyCs in visual cortex show a broad diversity of activation and suppression of activity during pupil dialation (*Stringer et al., 2016*; *Vinck et al., 2015*). The heterogeneity of ChC inhibition strength could be one circuit mechanism underlying this diversity. However, ChCs also receive input from local excitatory and inhibitory cells (*Figure 2—figure supplement 2*; *Jiang et al., 2015*; *Lu et al., 2017*), suggesting that their activity is also tuned by the local network state. The effect of richer sensory and behavioral context on ChC activity and synchrony will be an interesting question to explore in future work.

## A role for global inhibition in recurrent cortical circuits

The co-activation of ChCs that we observed in the physiology data and the lack of evidence of target specificity at the network level that we describe in the anatomy together suggest that ChCs deliver a common inhibitory signal to L2/3 excitatory cells. Each individual L2/3 excitatory cell receives more or less of that inhibitory signal and forms recurrent connections with other excitatory cells (*Dorkenwald et al., 2019*; *Lee et al., 2016*) as well as also targeting the ChCs. This connectivity motif, where interconnected excitatory network receives a common signal from a pool of inhibitory cells, who themselves are targeted by same local excitatory network, is a connectivity arrangement that resembles the soft winner-take-all (sWTA) motif originally proposed by *Amari, 1977* and others (*Douglas et al., 1995*; *Hahnloser et al., 2000*; *Maass, 2000*). Curiously enough Arbib was also one of the coauthors of the study that first described the ChCs (*Szentágothai and Arbib, 1974*). A basic feature of the sWTA motif is its ability to amplify the response of the subset of neurons that receive the strongest input,

while the responses of the others are suppressed due to the shared common inhibition, resulting in an increased signal-to-noise ratio of the excitatory cells. Similarly, our results show that ChCs are active during arousal states, which are known to be associated with an increase in the signal-to-noise ratio in pyramidal cells (*Vinck et al., 2015*).

## A framework to map cell-type connectivity rules

The underlying rules of connectivity between neurons depend not only on pre- and postsynaptic cell types, but geometry, morphology, and, as we have shown here, intrinsic properties of the cells involved. This work suggests that simple measures of connection strength like total synapse count have the potential to conflate cell-type factors with properties of the individual cells involved in a connection. Indeed, it is likely that connections between different cell types will depend in different ways on different cell-specific properties. Disentangling the multiple factors going into synaptic connections is a key aspect in understanding the development, maintenance, and function of cortical networks. Large-scale cortical connectomics (*Bock et al., 2011*; *Kasthuri et al., 2015*; *Lee et al., 2016*; *Motta et al., 2019*), particularly in the context of physiological measurements (*Bock et al., 2011*; *Lee et al., 2016*), offers the promise to examine the connections between identified cell types in the context of other structural, functional, and network measurements across a multitude of identified cells. The use of EM is crucial as it is the only way to densely map synaptic connectivity with concomitant measurements of detailed morphology and individual anatomical features for any cell. The connection from ChCs to L2/3 PyCs is a model case for investigating the nuances of cell-type-specific connectivity as the strong anatomical specificity makes a complete map easier to acquire. We anticipate that this approach, applied to the new generation of millimeter-scale EM volumes (*Bae et al., 2021*), will be a powerful tool to uncover wiring rules across the diversity of cortical cell types.

## Materials and methods

**Key resources table**

| Reagent type (species) or resource | Designation | Source or reference | Identifiers | Additional information |
|---|---|---|---|---|
| Antibody | Anti-GFP (chicken polyclonal) | Abcam | Cat# Ab13970; RRID:AB_300798 | (1:5000) |
| Antibody | Alexa-488 conjugated (donkey anti-chicken polyclonal) | Jackson ImmunoResearch | Cat# 703-545-155; RRID:AB_2340375 | (1:500) |
| Transfected construct (*Mus musculus*) | AAV1-CAG-FLEX-GCaMP6f | Addgene | Plasmid # 100835-AAV1; RRID:Addgene_100835 | *Chen et al., 2013* Used for ChC imaging |
| Transfected construct (*M. musculus*) | AAV5-CAG-FLEX-GCaMP6f | Addgene | Plasmid # 100835-AAV5; RRID:Addgene_100835 | *Chen et al., 2013* Used for ChC imaging |
| Transfected construct (*M. musculus*) | AAV9-CAG-FLEX-eGFP | Addgene | Plasmid # 51502-AAV9; RRID:Addgene_51502 | *Oh et al., 2014* Used for ChC imaging |
| Strain, strain background (*M. musculus*) | AI93 | The Jackson Lab | JAX Stock No. 024103; RRID:IMSR_JAX:024103 | *Madisen et al., 2015* Used for EM mouse |
| Strain, strain background (*M. musculus*) | Mouse: CamK2a-tTA/CamK2-Cre | The Jackson Laboratory | JAX Stock No. 003010; RRID:IMSR_JAX:003010 | *Mayford et al., 1996* Used for EM mouse |
| Strain, strain background (*M. musculus*) | Mouse: Vipr2-IRES2-Cre | The Jackson Laboratory | JAX Stock No. 031332; RRID:IMSR_JAX:031332 | *Daigle et al., 2018* Used for ChC imaging |

*Continued on next page*

*Continued*

| Reagent type (species) or resource | Designation | Source or reference | Identifiers | Additional information |
|---|---|---|---|---|
| Software, algorithm | Morphological and synapse analysis | This paper | | https://github.com/AllenInstitute/ ChandelierL23 (copy archived at swh:1:rev:f0087571f613eadf68cd6de0f93525a7ea949873, *Schneider-Mizell, 2021*) |
| Software, algorithm | Eye tracking | *Zhuang et al., 2017a, Zhuang, 2019* | https://github.com/ zhuangjun1981/eyetracker | Version 3.1; Used for ChC imaging |
| Software, algorithm | Two photo image preprocessing | *Zhuang et al., 2017a, Zhuang, 2017b* | https://github.com/ zhuangjun1981/stia/tree/ master/stia | Used for ChC imaging |
| Software, algorithm | Brain Modelling Toolkit | *Gratiy et al., 2018* | https://github.com/ AllenInstitute/bmtk | Version 0.0.7. Used for whole-cell modeling |

## Animal preparation for EM

All animal procedures were approved by the Institutional Animal Care and Use Committee at the Allen Institute for Brain Science or Baylor College of Medicine. Neurophysiology data acquisition was conducted at Baylor College of Medicine prior to EM imaging, but not used in this analysis. To aid in registration of optical physiology data to EM data, a widefield image of the cranial window visualizing the surface vasculature was provided in addition to a volumetric image stack of the vasculature, encompassing the region of tissue where the neurophysiology dataset was acquired. The vasculature was imaged by injecting a red fluorescent dye into the bloodstream of the mouse, allowing blood vessels and cell bodies to be imaged simultaneously by two-photon microscopy. Mice were then transferred to the Allen Institute in Seattle and kept in a quarantine facility for 1–3 days, after which they were euthanized and perfused.

## Mouse line for EM

The mouse used for EM was a triple-heterozygote for the following three genes: (1) Cre driver: CamKIIa_Cre (Jax:05359; https://www.jax.org/strain/005359), (2) tTA driver: B6;CBA-Tg(Camk2a-tTA)1Mmay/J (Jax: 003010; https://www.jax.org/strain/003010), and (3) GCaMP6f Reporter: Ai93(JAX 024103). It was perfused at P36.

## Perfusion

After induction of anesthesia with isoflurane, the appropriate plane of anesthesia was checked by a lack of toe pinch reflex and the animals were transcardially perfused with 15 ml 0.15 M cacodylate buffer (EMS, Hatfield, PA, pH 7.4) followed by 30 ml fixative mixture containing 0.08 M cacodylate (pH 7.4), 2.5% paraformaldehyde (PFA) (EMS), 1.25% glutaraldehyde (EMS), and 2 mM calcium chloride (Sigma). The perfusion solution was based on the work of *Hua et al., 2015*. Once the brain was removed, it was placed into the same fixative solution to post-fix for 16–72 hr at 4°C.

## Identifying neurophysiological region for further EM processing

To accurately identify and isolate the region of cortical tissue where the neurophysiology dataset was imaged, we labeled and imaged the surface and descending vasculature. After perfusion of the animals and excision of the brain, the surface of the cortex was imaged using differential contrast lighting to visualize the surface vasculature of visual cortex where the cranial window had previously been. We manually marked fiduciary points around this region of cortex to aid identification of the previously imaged cortical site after vibratome sectioning. The brain was washed in CB (0.1 M cacodylate buffer pH 7.4) and embedded in 2% agarose. The agarose was trimmed and mounted for coronal sectioning in a Leica VT1000S vibratome; successive 200-µm-thick slices were taken until the entire region of cortical tissue previously demarcated by manual markings was sectioned. During this procedure, we also acquired blockface images of each brain slice.

After vibratome sectioning, the widefield image of the cranial window and the vasculature stack were co-registered with the images of the surface vasculature from the brain surface using TrakEM2 (*Cardona et al., 2012*). From the blockface images, we could readily identify the fiduciary marks made on the brain surface. A volumetric representation of the cortical surface of the blockface images and the fiduciary marks was constructed in TrakEM2, and the orientation and position of the vibratome sections were aligned to the surface vasculature images by affine registration of the fiduciary points. This allowed for determination of the vibratome slices that contained the previously imaged region of cortical tissue.

To map the neurophysiology imaged site within the coronal slice, we next mounted the slices under coverglass in CB and acquired 10× images of the entire hemisphere of the slice, and 20× image stacks covering the cortical tissue surrounding the potential imaged site using a Zeiss AX10 ImagerM2 upright light microscope. These 10× and 20× stacks were co-registered in TrakEM2, and these images allowed us to visualize the descending vasculature within the coronal sections. We next generated a volumetric rendering of the vasculature stack provided by Baylor College of Medicine using microView (Parallax Innovations). From this rendering, we could reslice and visualize the descending vasculature from the previously imaged site. Corresponding vasculature landmarks were identified between the in vivo imaged site and the light microscopy coronal slice stacks. These landmarks were used to map the extent of the previously imaged tissue site to the coronal slice. The coronal sections containing the imaged site were then selected for histological processing (see below).

## EM histology

The histology protocol used here is based on the work of *Hua et al., 2015*, with modifications to accommodate different tissue block sizes and to improve tissue contrast for transmission electron microscopy (TEM).

Following several washes in CB (0.1 M cacodylate buffer pH 7.4), the vibratome slices were treated with a heavy metal staining protocol. Initial osmium fixation with 2% osmium tetroxide in CB for 90 min at room temperature was followed by immersion in 2.5% potassium ferricyanide in CB for 90 min at room temperature. After 2 × 30 min washes with deionized (DI) water, the tissue was treated with freshly made and filtered 1% aqueous thiocarbohydrazide at 40°C for 10 min. The samples were washed 2 × 30 min with DI water and treated again with 2% osmium tetroxide in water for 30 min at room temperature. Double washes in DI water for 30 min each were followed by immersion in 1% aqueous uranyl acetate overnight at 4°. The next morning, the samples in the same solution were placed in a heat block to raise the temperature to 50° for 2 hr. The samples were washed twice in DI water for 30 min each, then incubated in Walton's lead aspartate pH 5.0 for 2 hr at 50°C in the heat block. After double washes in DI water for 30 min each, the slices were dehydrated in an ascending ethanol series (50%, 70%, 90%, 3 × 100%) 10 min each and two transition fluid steps of 100% acetonitrile for 20 min each. Infiltration with acetonitrile:resin dilutions at 2p:1p (24 hr), 1p:1p (48 hr), and 1p:2p (24 hr) were performed on a gyratory shaker. Samples were placed in 100% resin for 24 hr, followed by embedment in Hard Plus resin (EMS, Hatfield, PA). The samples were cured in a 60°C oven for 96 hr.

In order to evaluate the quality of samples during protocol development and before preparation for large-scale sectioning, the following procedure was used for tissue mounting, sectioning, and imaging. We evaluated each sample for membrane integrity, overall contrast, and quality of ultrastructure. For general tissue evaluation, adjacent slices and tissue sections from the opposite hemisphere, processed in the same manner as the region of interest (ROI) slice, were cross-sectioned and thin sections were taken for evaluation of staining throughout the block neighboring the ROI.

## Ultrathin sectioning

The tissue block was trimmed to contain the neurophysiology recording site, which is the ROI, then sectioned to 40 nm ultrathin sections. For both trimming and sectioning, a Leica EM UC7 ultramicrotome was equipped with a diamond trimming tool and an Ultra 35 diamond knife (DiATOME USA), respectively. Sectioning speed was set to 0.3 mm/s. 8–10 serial thin sections were cut to form a ribbon, after which the microtome thickness setting was changed to 0 nm in order to release the ribbon from the knife edge. Then, using an eyelash superglued to a handle, ribbons were organized

to pairs and picked up as pairs to copper grids (Pelco, SynapTek, 1.5 mm slot hole) covered by 50-nm-thick LUXFilm support (Luxel Corp., Friday Harbor, WA).

## EM imaging

The imaging platform used for high-throughput serial section imaging is a JEOL-1200EXII 120 kV transmission electron microscope that has been modified with an extended column, a custom scintillator, and a large format sCMOS camera outfitted with a low distortion lens. The column extension and scintillator facilitate an estimated 10-fold magnification of the nominal field of view with negligible impact on resolution. Subsequent imaging of the scintillator with a high-resolution, large-format camera allows the capture of fields of view as large as 13 × 13 µm at 4 nm resolution. As with any magnification process, the electron density at the phosphor drops off as the column is extended. To mitigate the impact of reduced electron density on image quality (shot noise), a high-sensitivity sCMOS camera was selected and the scintillator composition tuned in order to generate high-quality EM images within exposure times of 90–200 ms (*Yin et al., 2020*).

## Proofreading and annotation of volumetric imagery data

We used a combination of Neuroglancer (RRID:SCR_015631) and custom tools to annotate and store labeled spatial points. In brief, we used Neuroglancer to simultaneously visualize the imagery and segmentation of the 3D EM data. A custom branch of Neuroglancer was developed that could interface with a 'dynamic' segmentation database, allowing users to correct errors (i.e., either merging or splitting neurons) in a centralized database from a web browser. Neuroglancer has some annotation functionality, allowing users to place simple annotations during a session, but does not offer a way to store them in a central location for analysis. We thus built a custom cloud-based database system to store arbitrary annotation data center associated with spatial points that could be propagated dynamically across proofreading events. Annotations were programmatically added to the database using a custom Python client and, in relevant cases, after parsing temporary Neuroglancer session states using custom Python scripts. These spatial points and their associated data (e.g., synapse type, cell body ID number, or cell types) were linked to stored snapshots of the proofreading for querying and reproducible data analysis. All data analyzed here came from the 'v183' snapshot.

## Visualization and analysis of mesh data

Neuronal meshes were computed by Igneous, version 0.1.0 (*Macrina et al., 2021*) (https://github.com/seung-lab/igneous) and kept up to date across proofreading. Meshes were analyzed in a custom Python library, MeshParty, (*Dorkenwald et al., 2020b*) version 1.14.0 (https://github.com/sdorkenw/MeshParty), that extends Trimesh (https://trimsh.org) with domain-specific features and VTK (https://www.vtk.org) integration for visualization. In cases where skeletons were used, we computed them with a custom modification of the TEASAR algorithm (*Sato et al., 2000*) on the vertex adjacency graph of the mesh object implemented as part of MeshParty. In order to associate annotations such as synapses or AIS boundary points with a mesh, we mapped point annotations to the closest mesh vertex after removing artifacts from the meshing process.

## AIS identification and extraction

To get a handle on the contents of the EM volume, we manually identified every cell body in the dataset manually and assigned a unique point at the approximate center of the soma for each. We then manually assigned a coarse cell type (excitatory, inhibitory, or glia) to each based on morphology. To identify the AIS of excitatory neurons, we manually placed points at the top and bottom of excitatory neurons that were thought to have a largely completely AIS in the volume. To extract the mesh vertices associated with the AIS, we computed the path distance between each mesh vertex and the top and bottom points ($d_{top}$ and $d_{bottom}$, respectively), the distance between top and bottom points ($d_{AIS}$), and kept those vertices that satisfied $d_{top}^2 + d_{bottom}^2 < \left(d_{AIS} + 1000\right)^2$. Distances were measured in nm using Dijkstra's algorithm as implemented in SciPy (https://www.scipy.org). The constant padding term allows the mesh definition to wrap around the AIS smoothly.

To extract the initial 37 microns of the AIS, we skeletonized each AIS (see previous section) and computed which mesh vertices were closest to each skeleton vertex. We filtered out all mesh vertices associated with the skeleton vertices more than 37 microns of the top skeleton tip. The threshold was

chosen to balance keeping as many distinct PyCs as possible while covering as much of the ChC input domain as possible.

## ChC type classification and proofreading

Cell typing of AIS inputs was performed across several rounds of proofreading and annotation. Every neurite presynaptic to an AIS was evaluated starting from its synapses. The main step was to evaluate other synapses from the same axon. The compartment (AIS, dendrite, or soma) was trivial to determine via manual inspection from the automated segmentation, even without labels or proofreading. If any of those synapses targeted dendrites or soma at subsequent points along the axon, the axon was labeled non-ChC at the seed synapses. Attaching the annotation to this point would allow any potential distant splitting of the axon due to proofreading to remain unlabeled. In contrast, axons that exhibited multi-bouton cartridges characteristic of ChCs and only targeted AISs were labeled as ChCs and proofread completely, extending tips and splitting segmentation errors. Because of the large number and size of these cells, comprehensive proofreading of non-ChC was beyond the scope of the project. However, to account for the possibility of ChC axons that had been erroneously merged into non-ChC axons by the automated segmentation, we evaluated every non-ChC synapse after completing an initial round of axon classification. Non-ChC axons forming multisynaptic contacts onto any AIS were also given additional scrutiny. A small number of axonal fragments that targeted AIS near the edge of the volume had few synapses overall and were more difficult to classify. In those cases, we used bouton morphology, tight clustering with established ChC boutons, and in some instances following the axonal process in imagery outside the segmented region to cartridges.

## Soma and AIS mesh structural properties

The automated meshing process introduced artifacts that made measuring spatial properties like surface area require special processing. For example, if part of the nucleus was segmented separately, this would introduce extra mesh faces to the neuron. In order to extract a clean surface for cell somata, we extracted a cutout of the voxel segmentation within 15 microns of the approximate cell body center that contained both the soma and initial part of proximal neurites. Boundary expansion and contraction for the segmentation was performed to fill small gaps caused by image and segmentation artifacts, and the mesh was recomputed using the marching cubes algorithm. We then used CGAL (https://www.cgal.org) surface mesh segmentation to identify proximal neurites and leave only the core soma mesh for measurements. Manual quality control of resulting meshes was done to ensure that the analysis only included cell bodies that were completely contained in the volume and were free of remaining artifacts. Surface area was computed by summing the area of mesh faces. Synapse count was computed from those synapses associated with the core soma mesh.

We computed AIS radius from the mesh via ray tracing. From each point on the AIS skeleton, a ray was sent toward the opposite side and the intersection with the opposite point was used to determine the diameter at that point. The AIS can emerge from either the soma or a dendrite, which could influence the radius of the most proximal part of the AIS due to the transition between compartments. To avoid being affected by transition, we ignored the initial 5 µm and averaged the radius based on skeleton vertices between 5 and 38 µm along the AIS.

The imagery dataset was sectioned so that the y-axis was approximately aligned with the pia-to-white-matter dimension, but exact alignment was not possible at the data collection stage. To more accurately measure depth within the data, we rotated the coordinate system to align with the average vector from AIS top to AIS bottom across all cells.

## Structural components analysis

We could compute complete AIS and soma features for 113 cells in the data. Most cells that were excluded had soma that touched the edge of the volume and thus were only partially reconstructed. Pearson correlations between structural features were found using SciPy with a Holm–Sidak multiple test correction. To address the underlying correlations, we used the FastICA implementation in scikit-learn to do a components analysis of the structural properties. We selected three components as the first three components of a PCA (an approximation and bound on ICA explained variance) account for 84% of the variance and three ICA components were clearly interpretable. FastICA is a stochastic

algorithm, so we ran it many times and selected the most robust solution. Components were multiplied by –1 if needed to make the largest element positive.

To look at the relationship between structural components and ChC input, we used OLS regression on z-scored counts of synaptic input, number of connections, and average synapses per connection. Coefficients and confidence intervals were computed with Statsmodels (https://www.statsmodels.org), and p-values were adjusted with a Holm–Sidak multiple test correction.

## Somatic synaptic input categorization

To check whether individual synapses onto pyramidal cell soma were excitatory or inhibitory, we randomly selected six pyramidal cells with full cell bodies for analysis. For each cell, we identified all synaptic inputs within 13 microns of the centroid of the cell body and excluded synapses onto the proximal part of dendritic branches or the AIS. For those synapses onto the soma, we looked at the presynaptic axon and checked 2–5 nearby boutons that synapse onto other excitatory cells. The presynaptic axon was classified as putatively inhibitory if the synapses were onto dendritic shafts, somata, or dendritic spines with more than one synaptic input and excitatory if synapses were principally onto dendritic spines. If an axon had no other boutons in the volume or all were onto inhibitory neurons, it was ignored for analysis. For five of the selected pyramidal cells (IDs: 648518346349534289, 648518346349537718, 648518346349539856, 648518346349538711, and 648518346349537516), a random sampling of 81 somatic synaptic inputs found all presynaptic axons were putatively inhibitory (N = 12, 16, 27, 13, and 13 synapses per cell). For one additional pyramidal cell (ID: 648518346349539768), we examined all somatic inputs (N = 75) and found that 100% were putatively inhibitory.

## Potential synapse analysis

We defined a potential connection as a ChC axon whose mesh came within a given distance threshold of a truncated AIS mesh (as described above). Distances were measured with the SciPy implementation of the k-d tree data structure. We tested distance thresholds between 5 and 15 µm and saw qualitatively similar results for 5–10 µm. Since the number of true connections was irrespective of distance, connectivity threshold was strictly nonincreasing with increased distance. For a distance threshold of 15 µm, connectivity fraction showed a negative trend with depth, suggesting that the influence of depth on potential connections became a factor at that length scale due to density of nearby axons, and we report only the results from the lower range of thresholds. For some AISs near the edge of the volume, the region around them might reach beyond the edge of the segmentation, resulting in a potential undercounting of potential connections. To avoid including these AIS in our analysis, we computed the fraction of voxels within the distance threshold that were within the segmented data. If more than 10% of the voxels were outside of the segmentation for a given distance threshold, the cell was omitted from that analysis.

## Network motif analysis

Based on the definition of potential connection above, we generated two bipartite networks from ChCs onto PyCs, one based on the potential connectivity, and one based on actual synaptic connectivity. By definition, the actual connectivity network is a strict subset of the potential network. To investigate the clumpiness of ChC targeting, we looked at the bifan motif in the bipartite network from ChC to PyCs. A bifan is defined as the motif comprising two source nodes (ChCs) and two target nodes (PyCs) and four edges, here such that two ChC target the same two PyCs. We generalized the concept to include the concept of potential synapse, defining a 'potential bifan' as a bifan where one edge was a potential connection and the other three were actual connections. For every potential connection, we evaluated if it was part of a potential bifan and then if it was an actual bifan. The bifan connectivity fraction was then the ratio of these two numbers.

We next developed a method to randomize the actual network within the potential network. Starting from the observed actual network, we iterated through each actual connection in a random order and at each step we (1) removed the actual connection from the graph, leaving it only a potential connection, and (2) for the target AIS, picked a new potential, but not already connected, ChC (including the just-disconnected one). Each step preserves the degree distribution of the AIS, but not the ChC, which we chose to reflect the apparent postsynaptic influence on ChC input and relative

uniformity of ChC axons. We iterate through the network five times, shuffling each edge each time. To add a bias to the shuffling, at the step of picking a new actual connection we evaluated which potential connections could complete a bifan motif. For each ChC connection , it was given a weight $w_i = \alpha$ if it was a potential bifan and $w_i = 1$ otherwise. A connection was then selected with probability proportional to its weight. The result is that for $\alpha > 1$, connectivity will be clumpier with bifans being generated more often than chance, while for $\alpha < 1$, connectivity will be more dispersed, with bifans being generated less often than chance.

## CO annotation and analysis

To annotate the COs, we selected 10 PyCs intentionally from across the distribution of total ChC synapse count. An expert neuroanatomist worked through every 3–4 sections of the data, placing annotation points on ER with stacked cisternae. Cisternae points were placed within the extent of the organelle every several sections, creating a point cloud throughout.

To compute the location of the COs, we mapped the stacked cisternae points to the closest mesh index on the AIS. This left both clear clumps associated with COs and a few diffuse outliers due to ambiguous ultrastructure and the mapping procedure. To filter the data down to just the clumps, we used a density-based clustering algorithm DBSCAN (citation) as implemented in scikit-learn, where the distance between points was computed along the mesh vertices and edges.

While synapse detection associated a location with every synapse, we found that there was a bias in this location away from *en face* parts of the synapse active zone and onto the transversely cut, resulting in an axis-aligned bias of AIS synapse locations. To more accurately assign a location to each synapse, we associated each synapse with the contact site between the presynaptic ChC mesh and the postsynaptic AIS mesh. The contact site was computed by finding nodes of the AIS mesh within 150 nm of the ChC mesh and clustering with DBSCAN (*Ester et al., 1996*) using precomputed distances along the mesh surface, which generated puncta-like clusters for each bouton contact. Each synapse could then be associated with a given puncta, and the center of the puncta was computed by identifying the node with the highest average distance along the mesh surface from other nodes within the puncta.

For each synapse, we computed its depth and orientation of the mesh vertex associated with it. Depth was measured as the distance along the skeleton from the top to the skeleton node closest to the synapse mesh vertex. Orientation was measured by computing a slice of the AIS mesh centered at the synapse and spanning 400 nm along the pia-to-white matter direction (the y-axis of the dataset). The AIS mesh points in this slice were projected into 2D, and their convex hull was used to estimate the local outline of the AIS. The angle of the vector from the center of the convex hull to the synapse was used for the orientation, with an orientation angle of 0 corresponding to the positive x-axis direction. To map vertices across AISes, we used their depth and orientation values of each synapse to compute the vertex on the target AIS with the most similar orientation that was also within 200 nm of the same depth.

## Generating biophysically detailed all-active models

The Allen Cell-Types database (http://celltypes.brain-map.org) contains 1920 in vitro whole-cell patch-clamp recordings and 485 biocytin-filled digital reconstructions from neurons in mouse primary visual cortex for a variety of transgenic lines. The all-active single-neuron models are constrained by these two data modalities from the same cell, namely, electrophysiology – voltage responses under standardized set of protocols and morphology – diameter and length of each segment within the tree. We distribute voltage-gated sodium (Na$^+$), potassium (K$^+$), and calcium (Ca$^{++}$) conductances across the entire morphology; specifically, we use the following channels Ih, NaT, NaP, KT, KP, Kv2, Kv3.1, SK, Im, CaLVA, CaHVA, with the assumption that these ion channels are expressed uniformly along each major morphological sections: soma, axon, apical, and basal dendrites. These parameters and the passive membrane properties (membrane capacitance cm, axial resistance Ra, leak conductance g_pas, reversal potential e_pas) construct the multicompartmental model of the neuron, thereby forming the variable vector (n = 43) for the optimization. To avoid inconsistencies in the axonal reconstructions such as isolated segments (or absence of axon altogether), we replace the axon with a 60-µm-long, 1-µm-diameter initial segment. We use a multiobjective optimization framework (*Druckmann et al., 2007*) where features such as action potential (AP) amplitude, width, spike frequency,

steady-state voltage, etc., are extracted from individual traces and the deviation (z-score) between experiment and model feature at a specific stimulus becomes one of the objectives. For our purposes, we have used Python toolbox BluePyOpt (*Van Geit et al., 2016*) that offers evolutionary algorithms (*Fortin et al., 2012*) to solving multiobjective optimization problems, with NEURON 7.5 (*Hines and Carnevale, 1997*) under the hood to simulate each model spawned out of the parameter explorations. To get a handle on the computation, we have designed a three-stage workflow where we progressively introduce new channels to the circuit, that is, first only fit the passive parameters with features from subthreshold experimental traces, next add hyperpolarization activated channel Ih with sag (*Hogan and Poroli, 2008*)-related features to the objective, and finally equip the circuit at its full complexity by introducing the rest of the channels and fit the conductance densities with both spiking and subthreshold trace features. Throughout the workflow, the Indicator-Based Evolutionary Algorithm (IBEA) adds 512 new offsprings (new models) to the population at each generation with Stage 0, 1 evolved up to 50 generations with one seed and Stage 2 continued for 200 generations with four independent seeds. On a 256 × 2.2 GHz Intel Xeon E5-2630v4 distributed cluster with 150 Gb of maximum process memory, the optimization of a single cell takes 26 ± 11 hr. Overall, we have used 15 distinct features across the three stages extracted with the eFEL library (*Van Geit, 2018*). Our models capture axonal AP initiation, an important aspect of biophysical realism. To add this constraint, we append the Boolean feature checkAISInitiation, part of the eFEL library, at the final stage of the optimization. This involves calculating the AP onset at axon and soma and adding a heavy penalty to the models for which somatic AP precedes axonal AP. This also requires allowing a less restrictive maximal density for the transient $Na^+$ current and lower action potential threshold at the axon compared to the soma. At the conclusion of the three stages, the workflow outputs 40 models sorted according to the sum of all objectives, for each experiment. We use the 'best model,' that is, the model with least training error for downstream analysis of synaptic integration properties.

## Simulating single-neuron models under synapses

For simulating the single-neuron models under synaptic inputs, we have used Brain Modeling Toolkit (bmtk) (*Gratiy et al., 2018*) with NEURON 7.5 simulation environment. In this study, we have used all-active models for four pyramidal cells (cortical layers 2–4) with IDs: 477127614, 571306690, 584254833, 382982932 from the Allen Cell-Types database. For *Figure 7E and F*, we have simulated each model for 1 s with 50 excitatory synapses over 25 connections (two synapses per connection) with Poisson spike train rates ranging from 0 to 500 Hz. The number of inhibitory synapses on the AIS of the modeled PyC and their distribution is adopted from EM data, and the frequency of the incoming inhibitory spike train is held constant at 100 Hz. For each cell and number of inhibitory synapses to evaluate the resultant 'fi curve' (e), we run eight different repetitions at each excitation rate. We aggregate the spike count output for these simulations and group them according to the amount of inhibition and perform one-sided pairwise Wilcoxon signed-rank test at 5% false discovery rate (FDR).

For the simulations in *Figure 7G*, both excitatory and inhibitory spike trains are sampled from a Gaussian rate function with the maximal rate of 500 Hz and 200 Hz, respectively. The peak of the excitation is varied at 5 ms intervals with 10 repetitions to capture the variability in response, and the total simulation window is 1 s. The number of excitatory synapses and their distribution (25 * 2) remains unchanged. For the pairwise comparison, we once again use one-sided Wilcoxon signed-rank test at 5% FDR. The free parameters in these simulations, such as the maximal excitation, inhibition rate, or number of excitatory synapses, are selected such that the cells represented by the biophysical models operate at a similar activation regime.

## Surgery and animal preparation for in vivo imaging of ChCs

In total, three Vipr2-IRES2-Cre mice (one male, two females) were used in this study. The surgery included a stereotaxic viral injection and a cranial window/head-plate implantation. During the injection, a glass pipette back-loaded with AAV virus was slowly lowered into the superficial layer of left V1 (3.8 mm posterior 2.7 mm lateral from bregma, 0.3 mm below pia) through a burr hole. 5 min after reaching the targeted location, 50 or 100 nl of virus was injected into the brain over 10 min by a hydraulic nanoliter injection system (Nanoject III, Drummond). The pipette then stayed for an additional 10 min before it was slowly retracted out of the brain. AAV1 (or AAV5)-CAG-FLEX-GCaMP6f

(Addgene: 100835-AAV1 or AAV5, titer 2 × 10$^{13}$ vg/ml) were used for functional imaging and AAV9-CAG-FLEX-eGFP (Addgene: 51502-AAV9, titer 2.28 × 10$^{13}$ vg/ml, 1:10 dilution) was used for structure imaging. Immediately after injection, a titanium head-plate and a 5 mm glass cranial window were implanted over left V1 following the Allen Institute standard procedure protocol (*de Vries et al., 2020*; for detailed protocol, see *Goldey et al., 2014*) allowing in vivo two-photon imaging during head fixation.

After surgery, the animals were allowed to recover for at least 5 days before retinotopic imaging with intrinsic signal during anesthesia (for detailed retinotopic protocol, see *Juavinett et al., 2017*). After retinotopic mapping, animals were handled and habituated to the imaging rig for two additional weeks before in vivo two-photon imaging (*de Vries et al., 2020*).

## Histology for viral expression

To characterize the Cre expression pattern, Vipr2-IRES2-Cre mice injected by Cre-dependent eGFP or GCaMP viruses were perfused and brains collected. Briefly, mice were anesthetized with 5% isoflurane and 10 ml of saline (0.9% NaCl) followed by 50 ml of freshly prepared 4% PFA was pumped intracardially at a flow rate of 9 ml/min. Brains were immediately dissected and post-fixed in 4% PFA at room temperature for 3–6 hr and then overnight at 4°C. After fixation, brains were incubated in 10% and then 30% sucrose in PBS for 12–24 hr at 4°C before being cut into 50 μm sections by a freezing-sliding microtome (Leica SM 2101R). Sections from V1 were mounted on gelatin-coated slides and cover-slipped with Prolong Diamond Antifade Mounting Media (P36965, Thermo Fisher). For GCaMP-labeled tissue, sections were processed with antibody staining before mounting. During antibody staining, sections containing LGN and V1 were blocked with 5% normal donkey serum and 0.2% Triton X-100 in PBS for 1 hr, incubated in an anti-GFP primary antibody (1:5000 diluted in the blocking solution, Abcam, Ab13970) for 48–72 hr at 4°C, washed the following day in 0.2% Triton X-100 in PBS and incubated in a Alexa-488 conjugated secondary antibody (1:500, 703-545-155, Jackson ImmunoResearch) and DAPI.

The sections were then imaged with Zeiss AxioImager M2 widefield microscope with a 10×/0.3 NA objective. Fluorescence from antibody enhanced GCaMP and mRuby3 were extracted from filter sets Semrock GFP-1828A (excitation 482/18 nm, emission 520/28 nm, dichroic cutoff 495 nm) and Zeiss # 20 (excitation 546/12 nm, emission 608/32 nm, dichroic cutoff 560 nm), respectively. A subset of sections was imaged with two-photon microscope (Scientifica 2PIMS, objective: Nikon 16XLWD-PF, 16×/0.8 NA, laser: Coherent Chameleon Ultra II, excitation wavelength: 920 nm, emission filter: 470–558 nm band-pass) to obtain 3D axon morphology with high resolution (pixel size: 0.47 μm).

## In vivo two-photon imaging

When recovery, retinotopic mapping, and habituation were finished (usually more than 3 weeks after initial surgery), V1 cells labeled with GCaMP6f were evident in superficial layers (50–200 μm) through the cranial window. Calcium activities from those cells were imaged by two-photon excitation using a custom microscope and 940 nm illumination by a Ti:sapphire laser (a Spectra-Physic Insight X3), focused with a 16×/0.8 NA objective (Nikon N16XLWD-PF). This scope has the ability to correct optical aberration (adaptive optics) and quickly switch focal depth by modulating the beam wavefront with a liquid crystal spatial light modulator (SLM, Meadowlark Optics, HSP-512, *Liu et al., 2018*). With this scope, we recorded calcium activities from planes at three different depths (16 or 32 μm apart in depth) in single imaging sessions. The plane at each depth was sequentially imaged at about 37 Hz and the volume rate was about 12 Hz, and aberrations out of the objective were corrected (with 512 × 512 pixels resolution at LSM). Emitted light was first split by a 735-nm dichroic mirror (FF735-DiO1, Semrock). The short-wavelength light was filtered by a 750-nm short-pass filter (FESH0750, Thorlabs) and a 470–588-nm bandpass emission filter (FF01-514/44-25, Semrock) before collected as GCaMP signal. Image acquisition was controlled using Vidrio Scan-Image software (*Pologruto et al., 2003*). To maintain constant immersion of the objective, we used gel immersion (Genteal Gel, Alcon).

All imaging sessions were performed during head fixation with the standard Allen Institute Brain Observatory in vivo imaging stage (*de Vries et al., 2020*).

## Two-photon image preprocessing

The recorded two-photon movies for each imaging plane were motion-corrected using rigid body transform based on phase correlation by a custom-written Python package (*Zhuang et al., 2017a*; *Zhuang, 2017b*). To generate ROIs for boutons, the motion-corrected movies were further downsampled by a factor of 3 and then processed with constrained non-negative matrix factorization (CNMF) (*Pnevmatikakis et al., 2016*), implemented in the CaImAn Python library (*Giovannucci et al., 2019*). These ROIs were further filtered by their size (ROIs smaller than 23.4 µm$^2$ or larger than 467.5 µm$^2$ were excluded) and position (ROIs within the motion artifacts were excluded). Since the labeled cells distributed sparsely, there were no overlapping pixels among ROIs. For each retained ROI, a neuropil ROI was created as the region between two contours by dilating the ROI's outer border by 1 and 8 pixels excluding the pixels within the union of all ROIs. The calcium trace for each ROI was calculated by the mean of pixelwise product between the ROI and each frame of the movie, and its neuropil trace was calculated in the same way using its neuropil ROI. To remove the neuropil contamination, the neuropil contribution of each ROI's calcium trace was estimated by a linear model and optimized by gradient descendent regression with a smoothness regularization (*Zhuang et al., 2017a*; *de Vries et al., 2020*).

## Pupil area and locomotion speed extraction

During each imaging session, the locomotion speed and a movie of the animal's right eye were simultaneously recorded following the Allen Brain Observatory standard protocol (*de Vries et al., 2020*; *Zhuang et al., 2017a*). To extract pupil area, the contour of pupil in each frame was extracted with adaptive thresholding (*Zhuang et al., 2017a*) by custom-written Python package 'eyetracker' version 3.1 (*Zhuang, 2019*) .

For each imaging session, a comprehensive collection of data including metadata, visual stimuli, all preprocessing results, final calcium traces, locomotion speed, and pupil area was generated in Neurodata Without Borders (nwb) 1.0 format with 'ainwb' package, version 1.0.2 (*Keithg and Nicain, 2018*). The analysis pipeline (imaging preprocessing, pupil/locomotion analysis, and nwb packaging) was performed by a custom-written Python package 'corticalmapping' version 2.0.0 (*Zhuang, 2014*).

## Correlation analysis

We computed pairwise correlations between the activities of all recorded cells for each imaging session during spontaneous periods. For the ChC recordings, we analyzed cells across the three different imaging planes. For the non-ChC types (VIP, SST, and PV), we used data recorded from a single-imaging plane in visual area VISp. These data were downloaded through the publicly available Allen Brain Observatory using the AllenSDK (0.16.3). We first downsampled the corrected fluorescence traces from 30 Hz to 12 Hz (using the SciPy resample function) to match the sampling rate of the ChC recordings. We show the distribution of correlation coefficients across all simultaneously recorded cell pairs separated by cell type and report summary statistics comparing the mean correlation coefficients by cell type. We compute statistical significance between the chandelier and non-ChC correlation values using the non-parametric Mann–Whitney U test.

To compute correlations between cell activity and other behavioral covariates, for each cell, we computed the correlation between its calcium activity and pupil area/locomotion speed. We compute statistical significance between the mean correlation values for running and pupil diameter using the Wilcoxon signed-rank test.

## Acknowledgements

We thank Wenjing Yin for reimaging of sections with the EM. We thank Rob Young for managing the stitching and alignment pipeline at the Allen Institute for Brain Science (AIBS). We thank John Philips, Sill Coulter, and the Program Management team at the AIBS for their guidance for project strategy and operations. We thank Hongkui Zeng, Ed Lein, Christof Koch, and Allan Jones for their support and leadership. We thank the Manufacturing and Processing Engineering team at the AIBS for their help in implementing the EM imaging and sectioning pipeline. We thank Brian Youngstrom, Stuart Kendrick, and the Allen Institute IT team for support with infrastructure, data management, and data transfer. We thank the Facilities, Finance, and Legal teams at the AIBS for their support on the MICrONS

contract. We thank the Neurosurgery and Behavior and the Transgenic Colony Management teams at the AIBS for the preparation of mice for calcium imaging of chandelier cells. We thank Stephan Saal-feld for help with the parameters for 2D stitching and rough alignment of the dataset. We would like to thank the 'Connectomics at Google' team for developing Neuroglancer and computational resource donations. We also would like to thank Amazon and Intel for their assistance. We thank S Koolman, M Moore, S Morejohn, B Silverman, K Willie, and R Willie for their image analyses, Garrett McGrath for computer system administration, and May Husseini and Larry and Janet Jackel for project administration. We thank Ueli Rutishauser, Ahmed El Hady, and G Ocker for advice and feedback. Supported by the Intelligence Advanced Research Projects Activity (IARPA) via Department of Interior/Interior Business Center (DoI/IBC) contract numbers D16PC00003, D16PC00004, and D16PC0005. The U.S. Government is authorized to reproduce and distribute reprints for governmental purposes notwith-standing any copyright annotation thereon. *Disclaimer*: The views and conclusions contained herein are those of the authors and should not be interpreted as necessarily representing the official poli-cies or endorsements, either expressed or implied, of IARPA, DoI/IBC, or the U.S. Government. HSS also acknowledges support from NIH/NINDS U19 NS104648, ARO W911NF-12-1-0594, NIH/NEI R01 EY027036, NIH/NIMH U01 MH114824, NIH/NINDS R01NS104926, NIH/NIMH RF1MH117815, and the Mathers Foundation. We thank the Allen Institute for Brain Science founder, Paul G Allen, for his vision, encouragement, and support.

# Additional information

## Competing interests

Thomas Macrina: discloses financial interests in Zetta AI LLC. Jacob Reimer, Andreas S Tolias: discloses financial interests in Vathes LLC. H Sebastian Seung: discloses financial interests in Zetta AI. The other authors declare that no competing interests exist.

## Funding

| Funder | Grant reference number | Author |
|--------|------------------------|--------|
| Intelligence Advanced Research Projects Activity | D16PC00003 | Casey M Schneider-Mizell<br>Agnes L Bodor<br>Forrest Collman<br>Derrick Brittain<br>Adam Bleckert<br>Sven Dorkenwald<br>Nicholas L Turner<br>Thomas Macrina<br>Kisuk Lee<br>Ran Lu<br>Jingpeng Wu<br>Jun Zhuang<br>Anirban Nandi<br>Brian Hu<br>JoAnn Buchanan<br>Marc M Takeno<br>Russel Torres<br>Gayathri Mahalingam<br>Daniel J Bumbarger<br>Yang Li<br>Thomas Chartrand<br>Nico Kemnitz<br>William M Silversmith<br>Dodam Ih<br>Jonathan Zung<br>Aleksandar Zlateski<br>Ignacio Tartavull<br>Sergiy Popovych<br>William Wong<br>Manuel Castro<br>Chris S Jordan<br>Emmanouil Froudarakis<br>Lynne Becker<br>Shelby Suckow<br>Jacob Reimer<br>Andreas S Tolias<br>Costas A Anastassiou<br>H Sebastian Seung<br>R Clay Reid<br>Nuno Maçarico da Costa |

| Funder | Grant reference number | Author |
|---|---|---|
| Intelligence Advanced Research Projects Activity | D16PC00004 | Casey M Schneider-Mizell<br>Agnes L Bodor<br>Forrest Collman<br>Derrick Brittain<br>Adam Bleckert<br>Sven Dorkenwald<br>Nicholas L Turner<br>Thomas Macrina<br>Kisuk Lee<br>Ran Lu<br>Jingpeng Wu<br>Jun Zhuang<br>Anirban Nandi<br>Brian Hu<br>JoAnn Buchanan<br>Marc M Takeno<br>Russel Torres<br>Gayathri Mahalingam<br>Daniel J Bumbarger<br>Yang Li<br>Thomas Chartrand<br>Nico Kemnitz<br>William M Silversmith<br>Dodam Ih<br>Jonathan Zung<br>Aleksandar Zlateski<br>Ignacio Tartavull<br>Sergiy Popovych<br>William Wong<br>Manuel Castro<br>Chris S Jordan<br>Emmanouil Froudarakis<br>Lynne Becker<br>Shelby Suckow<br>Jacob Reimer<br>Andreas S Tolias<br>Costas A Anastassiou<br>H Sebastian Seung<br>R Clay Reid<br>Nuno Maçarico da Costa |

| Funder | Grant reference number | Author |
|---|---|---|
| Intelligence Advanced Research Projects Activity | D16PC0005 | Casey M Schneider-Mizell<br>Agnes L Bodor<br>Forrest Collman<br>Derrick Brittain<br>Adam Bleckert<br>Sven Dorkenwald<br>Nicholas L Turner<br>Thomas Macrina<br>Kisuk Lee<br>Ran Lu<br>Jingpeng Wu<br>Jun Zhuang<br>Anirban Nandi<br>Brian Hu<br>JoAnn Buchanan<br>Marc M Takeno<br>Russel Torres<br>Gayathri Mahalingam<br>Daniel J Bumbarger<br>Yang Li<br>Thomas Chartrand<br>Nico Kemnitz<br>William M Silversmith<br>Dodam Ih<br>Jonathan Zung<br>Aleksandar Zlateski<br>Ignacio Tartavull<br>Sergiy Popovych<br>William Wong<br>Manuel Castro<br>Chris S Jordan<br>Emmanouil Froudarakis<br>Lynne Becker<br>Shelby Suckow<br>Jacob Reimer<br>Andreas S Tolias<br>Costas A Anastassiou<br>H Sebastian Seung<br>R Clay Reid<br>Nuno Maçarico da Costa |
| National Institute of Neurological Disorders and Stroke | U19 NS104648 | H Sebastian Seung |
| Army Research Office | W911NF-12-1-0594 | H Sebastian Seung |
| National Eye Institute | R01 EY027036 | H Sebastian Seung |
| National Institute of Mental Health | U01 MH114824 | H Sebastian Seung |
| National Institute of Neurological Disorders and Strokescience | R01 NS104926 | H Sebastian Seung |
| National Institute of Mental Health | RF1MH117815 | H Sebastian Seung |
| Mathers Foundation | | H Sebastian Seung |

The funders had no role in study design, data collection and interpretation, or the decision to submit the work for publication.

## Author contributions

Casey M Schneider-Mizell, Gayathri Mahalingam, Conceptualization, Data curation, Formal analysis, Investigation, Software, Visualization, Writing – original draft, Writing – review and editing; Agnes L Bodor, Conceptualization, Data curation, Investigation; Forrest Collman, Conceptualization, Data curation, Formal analysis, Software, Visualization, Writing – original draft, Writing – review and editing; Derrick Brittain, Nicholas L Turner, JoAnn Buchanan, Dodam Ih, Investigation; Adam Bleckert, Thomas

Macrina, Jingpeng Wu, Yang Li, Data curation, Investigation, Software; Sven Dorkenwald, Data curation, Formal analysis, Investigation, Software; Kisuk Lee, Ran Lu, Marc M Takeno, Jonathan Zung, Aleksandar Zlateski, Ignacio Tartavull, William Wong, Manuel Castro, Investigation, Software; Jun Zhuang, Formal analysis, Investigation, Software, Writing – review and editing; Anirban Nandi, Brian Hu, Formal analysis, Investigation; Russel Torres, Funding acquisition, Project administration, Software, Supervision, Writing – original draft, Writing – review and editing; Daniel J Bumbarger, Sergiy Popovych, Data curation, Investigation, Resources; Thomas Chartrand, Formal analysis, Investigation, Writing – review and editing; Nico Kemnitz, Formal analysis, Investigation, Software; William M Silversmith, Formal analysis, Software; Chris S Jordan, Project administration, Software; Emmanouil Froudarakis, Lynne Becker, Investigation, Project administration; Shelby Suckow, Funding acquisition, Project administration, Supervision; Jacob Reimer, Conceptualization, Investigation, Supervision, Writing – original draft, Writing – review and editing; Andreas S Tolias, Conceptualization, Funding acquisition, Project administration, Supervision, Writing – original draft, Writing – review and editing; Costas A Anastassiou, Nuno Maçarico da Costa, Conceptualization, Data curation, Funding acquisition, Investigation, Project administration, Supervision, Writing – original draft, Writing – review and editing; H Sebastian Seung, Funding acquisition, Investigation, Project administration, Supervision, Writing – original draft, Writing – review and editing; R Clay Reid, Conceptualization, Funding acquisition, Project administration, Software, Supervision, Writing – original draft, Writing – review and editing

**Author ORCIDs**
Casey M Schneider-Mizell ⬤ http://orcid.org/0000-0001-9477-3853
Forrest Collman ⬤ http://orcid.org/0000-0002-0280-7022
Brian Hu ⬤ http://orcid.org/0000-0002-5866-8762
Marc M Takeno ⬤ http://orcid.org/0000-0002-8384-7500
Thomas Chartrand ⬤ http://orcid.org/0000-0002-7093-8681
Emmanouil Froudarakis ⬤ http://orcid.org/0000-0002-3249-3845
Andreas S Tolias ⬤ http://orcid.org/0000-0002-4305-6376
R Clay Reid ⬤ http://orcid.org/0000-0002-8697-6797
Nuno Maçarico da Costa ⬤ http://orcid.org/0000-0003-2001-4568

**Ethics**
All animal procedures were approved by the Institutional Animal Care and Use Committee at the Allen Institute for Brain Science (1503 and 1804) or Baylor College of Medicine (AN-4703).

**Decision letter and Author response**
Decision letter https://doi.org/10.7554/eLife.73783.sa1
Author response https://doi.org/10.7554/eLife.73783.sa2

---

## Additional files

**Supplementary files**
• Supplementary file 1. A .csv file with a list of links to interactively explore the electron microscopy (EM) data and segmentation in Neuroglancer. For links showing the input to a single axon initial segment (AIS), the number of synapses from chandelier cell (ChC) and non-ChC sources is also provided. Note that cell IDs are 64-bit integers and will not display correctly in some programs.

• Supplementary file 2. The supplementary file contains two .csv tables collecting the structural properties analyzed here. The first table contains information about all axon initial segment (AIS) synapses and the second contains all AIS and somatic properties. An included readme file describes the files in detail. Note that cell IDs are 64-bit integers and will not display correctly in some programs.

• Transparent reporting form

**Data availability**
Volume electron microscopy and segmentation data is available at https://www.microns-explorer.org/phase1.(https://doi.org/10.5281/zenodo.5646567). AIS synapse and PyC structural data are included in Supplemental Data. All other meshes and data tables are on Zenodo (https://doi.org/10.5281/zenodo.5579388). All ChC calcium traces are also on Zenodo (https://doi.org/10.5281/zenodo.5725826). Analysis code and NEURON models are available at http://github.com/AllenInstitute/

ChandelierL23 (copy archived at https://archive.softwareheritage.org/swh:1:rev:f0087571f613eadf68cd6de0f93525a7ea949873).

The following dataset was generated:

| Author(s) | Year | Dataset title | Dataset URL | Database and Identifier |
|-----------|------|---------------|-------------|------------------------|
| Becker L | 2020 | MICrONS Layer 2/3 Data Tables | https://zenodo.org/record/5579388 | Zenodo, 10.5281/zenodo.5579388 |
| Zhuang J | 2021 | Structure and Function of Axo-axonic Inhibition - Calcium Traces | https://zenodo.org/record/5725828 | Zenodo, 10.5281/zenodo.5725828 |

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
