## [Editor Report]

This paper will be of high interest to a broad audience of neuroscientists as it provides a major advancement of our understanding of cortical circuits. The quality and quantitative nature of the neuroanatomical reconstructions at synaptic resolution are remarkable. Complementing the reconstructions with computational modeling and activity measurements, the study proposes a likely circuit function for a specific inhibitory cell type during behavior.

---

## [Decision Letter]

**Decision letter after peer review:**

Thank you for submitting your article "The structural and functional logic of axo-axonic inhibition" for consideration by *eLife*. Your article has been reviewed by 3 peer reviewers, and the evaluation has been overseen by Ronald Calabrese as the Senior and Reviewing Editor. The following individuals involved in review of your submission have agreed to reveal their identity: Ed Callaway (Reviewer #1); Z. Josh Huang (Reviewer #2); Marcel Oberlaender (Reviewer #3).

Essential revisions:

No new experiments or analyses are required.

1) The P36 mouse visual cortex may not be fully mature and the authors should thoroughly discuss this possibility as outlined in the recommendations to authors of Reviewer #2.

2) Please address all the recommendations for the authors.

*Reviewer #2 (Recommendations for the authors):*

1. It is puzzling why the authors used a P36 mouse. For such a demanding and effort intensive work, they should have made sure that they would be studying the "end product" of ChC-PyN connectivity. They cited Inan et al., 2013 for evidence of "mature ChCs" at P36, but Inan et al., did not study older mice. We have unpublished evidence from high resolution complete single cell reconstruction that ChC axons and synapse "cartridges" continue to increase in complexity even at 2 months of age in the frontal cortex in mice; this timeline may also apply to visual cortex. If this were the case, then some of the results may pertain to a developing rather mature state of ChC -PyN connectivity. For example, the sparse non-ChC synapses at AIS may be further reduced or eliminated. I am not suggesting the authors to do another mouse at P60 for this paper, but this should be done at some point. For now, the authors should discuss the possibility that their results may not be describing the mature ChC circuit, and certain result could be pertaining to postnatal maturation.

2. The word "logic" is title is not quite warranted and should be replaced with a more descriptive term.

3. Line 226 "we conclude that axo-axonic ChC input along PyCs exhibites substantially higher variability than perisomatic inhibitory input". I think this is a very important result, which should be highlighted in Abstract, and Introduction or Discussion.

4. Line 38 in abstract, "using a targeted optogenetic approach" is misleading. "Optognetic approach" usually refers to optogenetic manipulation with opsins. However, the data shown here is two-photon calcium imaging of ChCs with Cre-dependent expression of GCaMP. Please consider modifying the description to make it accurate.

5. Line 208-254. "Chandelier cell input or ChC input to PyC" is a bit misleading. Please consider modifying. "ChC connections onto PyC" may be more accurate.

6. Are all ChC axon branches (122) from the 2 ChCs identified in current cubic tissue? Even though discussed in line 516-522, the assumption that "every ChC axon targeting the same AIS came from a different ChC" is not necessary. It would be more accurate to describe as "ChC axon branches" which is actually what is observed and the quantification is based on rather than ChC cells. The variability between different axon branches may suggest a heterogeneity among different axon branches from the same ChC.

7. Line 501. Does the description "Somatic input to PyCs is typically GABAergic" mean "axons targeting soma are mainly GABAergic"? Is there any excitatory synapse onto the soma of PyCs? If so, what is the proportion of excitatory and inhibitory synapses onto the soma of PyCs within current tissue?

8. line 555 Discussion paragraph on "A role for global inhibition", especially line 556-564. The authors suggest that "ChCs deliver a common inhibitory signal to L2/3 excitatory cells". This statement does not consider the possibility that subsets of PyNs distinguished by projection targets may be differentially innervated by ChCs. Indeed, this was shown by Lu et al., 2017 Nature Neuroscience, and ChCs in turn preferentially receive excitatory inputs from subsets of projection defined PyNs. The authors should discuss this work and not rush into a conclusion without the appropriate resolution for PyN projection subtypes.

*Reviewer #3 (Recommendations for the authors):*

The manuscript is well written, and the figures provide a clear account of the data. The reconstructions are spectacular and nicely complemented with simulation and functional imaging data. I have no suggestions that could improve the manuscript.

---

## [Author Response]

Reviewer #2 (Recommendations for the authors):1. It is puzzling why the authors used a P36 mouse. For such a demanding and effort intensive work, they should have made sure that they would be studying the "end product" of ChC-PyN connectivity. They cited Inan et al., 2013 for evidence of "mature ChCs" at P36, but Inan et al., did not study older mice. We have unpublished evidence from high resolution complete single cell reconstruction that ChC axons and synapse "cartridges" continue to increase in complexity even at 2 months of age in the frontal cortex in mice; this timeline may also apply to visual cortex. If this were the case, then some of the results may pertain to a developing rather mature state of ChC -PyN connectivity. For example, the sparse non-ChC synapses at AIS may be further reduced or eliminated. I am not suggesting the authors to do another mouse at P60 for this paper, but this should be done at some point. For now, the authors should discuss the possibility that their results may not be describing the mature ChC circuit, and certain result could be pertaining to postnatal maturation.

Following the reviewer suggestion, we have added a new section to the Discussion (“Development state of Chandelier connectivity”, Lines 639–672) where we discuss the possibility that the P36 might not reflect the final state of postnatal development. In this section, we also show two rare cases of filopodia observed on two chandelier axons, consistent with the idea that chandelier axonal arbors are still changing. In addition, in the Discussion section regarding single bouton connections (Lines 553–560), we raise the point that the abundance of single synapse connections may be a transient state en route to more developed, mature cartridges.

We should also note that the dataset for this paper was collected as one part of a large, collaborative effort and the mouse was as old as possible while still achieving the overall project requirements. The reviewer will be pleased to note that the project’s even more demanding and effort-intensive follow up dataset, the cubic millimeter of visual cortex (https://www.microns-explorer.org/cortical-mm3 — which so far has not been used to study chandelier cells), was taken from a P87 mouse.

2. The word "logic" is title is not quite warranted and should be replaced with a more descriptive term.

We have modified the title to “Structure and function of axo-axonic inhibition.”

3. Line 226 "we conclude that axo-axonic ChC input along PyCs exhibites substantially higher variability than perisomatic inhibitory input". I think this is a very important result, which should be highlighted in Abstract, and Introduction or Discussion.

We’ve highlighted the result about somatic input by mentioning it in the Abstract, Introduction and Discussion, as suggested. In addition, we have added the figure panel showing the distribution of somatic input synapses to the main Figure 3D.

4. Line 38 in abstract, "using a targeted optogenetic approach" is misleading. "Optognetic approach" usually refers to optogenetic manipulation with opsins. However, the data shown here is two-photon calcium imaging of ChCs with Cre-dependent expression of GCaMP. Please consider modifying the description to make it accurate.

The same issue was noted in the reviewer comment R1.1. We modified the abstract to be more specific. The abstract now reads “…using a cell-type specific calcium imaging approach…”.

5. Line 208-254. "Chandelier cell input or ChC input to PyC" is a bit misleading. Please consider modifying. "ChC connections onto PyC" may be more accurate.

Inspired by the reviewer comments, we looked at all uses of the term “input” in the paper. In cases where we intended to specifically refer to “ChC synapses onto PyC” or “ChC connections onto PyCs,” we clarified the wording to be more specific. This included considerable changes to the section noted in R2.5 (New line numbers). We believe that these changes more accurately convey our results.

6. Are all ChC axon branches (122) from the 2 ChCs identified in current cubic tissue? Even though discussed in line 516-522, the assumption that "every ChC axon targeting the same AIS came from a different ChC" is not necessary. It would be more accurate to describe as "ChC axon branches" which is actually what is observed and the quantification is based on rather than ChC cells. The variability between different axon branches may suggest a heterogeneity among different axon branches from the same ChC.

Not all the chandelier axon branches (122) are from the 2 ChCs identified in the volume. The volume is not large enough to encompass the entire axonal field of the ChC cells, and most of the branches in the volume are likely from other ChC cells whose soma are not in the volume. We have made this discussion more explicit (Lines 218–220). In addition, we have altered the Discussion section relating to the effect of volume truncation to center the quantification we performed (e.g. Line 354). We agree that describing these as ‘axon branches’ is more accurate and have changed the text to reflect this.

“Second, most ChC axon branches exit the volume, and we cannot tell which of these are part of the same axonal arbors. The number of distinct axon branch connections is thus an upper bound on the number of distinct presynaptic ChCs, although light-level morphology suggests that it is rare for multiple ChC axonal branches to target the same AIS. It is thus possible that not all conclusions made at the level of axon branches will translate directly to input at the level of individual ChCs.”

7. Line 501. Does the description "Somatic input to PyCs is typically GABAergic" mean "axons targeting soma are mainly GABAergic"? Is there any excitatory synapse onto the soma of PyCs? If so, what is the proportion of excitatory and inhibitory synapses onto the soma of PyCs within current tissue?

We have modified the text in the manuscript to clarify that we meant "axons targeting soma are mainly GABAergic". We have added citations (Lines 277-279) to additional ultrastructural analysis that suggest that most somatic input onto excitatory cells is inhibitory. In addition, we also measured this in our own data. We randomly sampled 156 synapses formed with the soma of six pyramidal cells (Lines 279–281 in main text and Methods section “Somatic Synaptic Input Categorization”). We sampled randomly from five cells and fully examined all synapses on the sixth. We found that all 156 somatic synapses were from axons whose excitatory targets were principally shafts, somata, or second synapses onto excitatory spines and thus are putatively GABAergic. We thus believe the number of excitatory inputs onto excitatory somata is negligibly small in this dataset.

8. line 555 Discussion paragraph on "A role for global inhibition", especially line 556-564. The authors suggest that "ChCs deliver a common inhibitory signal to L2/3 excitatory cells". This statement does not consider the possibility that subsets of PyNs distinguished by projection targets may be differentially innervated by ChCs. Indeed, this was shown by Lu et al., 2017 Nature Neuroscience, and ChCs in turn preferentially receive excitatory inputs from subsets of projection defined PyNs. The authors should discuss this work and not rush into a conclusion without the appropriate resolution for PyN projection subtypes.

We did not intend the statement to mean that each cell received the same amount of input, but rather that all ChCs appear to be mostly co-active and therefore carry the same “common signal”. That individual PyNs can receive more or less of that signal is indeed one of the major conclusions of this paper, which is consistent with Lu et al., 2017 (which we refer to in the section “Comparison with previous measures of ChC connectivity”). We have clarified the language to emphasize the distinction between uniform state-dependent ChC activity and the non-uniform connectivity from the ChC population onto PyCs (Lines 710–714).

“The co-activation of ChCs that we observed in the physiology data and the lack of evidence of target specificity at the network level that we describe in the anatomy together suggest that ChCs deliver a common inhibitory signal to L2/3 excitatory cells. Each individual L2/3 excitatory cell receives more or less of that inhibitory signal and forms recurrent connections with other excitatory cells (Dorkenwald et al., 2019, Lee et al., 2016) as well as also targeting the ChCs.”